# REGIONREASONER: REGION-GROUNDED MULTI-ROUND VISUAL REASONING

**Wenfang Sun**[*,1]    **Hao Chen**[*,2]    **Yingjun Du**[1]    **Yefeng Zheng**[†,3]    **Cees G. M. Snoek**[1]

[1]University of Amsterdam    [2]Anhui University    [3]Westlake University
[*]Equal contribution.    [†]Corresponding author.

## ABSTRACT

Large vision-language models have achieved remarkable progress in visual reasoning, yet most existing systems rely on single-step or text-only reasoning, limiting their ability to iteratively refine understanding across multiple visual contexts. To address this limitation, we introduce a new *multi-round visual reasoning* benchmark with training and test sets spanning both detection and segmentation tasks, enabling systematic evaluation under iterative reasoning scenarios. We further propose **RegionReasoner**, a reinforcement learning framework that enforces *grounded reasoning* by requiring each reasoning trace to explicitly cite the corresponding reference bounding boxes, while maintaining semantic coherence via a *global–local consistency reward*. This reward extracts key objects and nouns from both global scene captions and region-level captions, aligning them with the reasoning trace to ensure consistency across reasoning steps. RegionReasoner is optimized with structured rewards combining grounding fidelity and global–local semantic alignment. Experiments on detection and segmentation tasks show that *RegionReasoner-7B*, together with our newly introduced benchmark **RegionDial-Bench**, considerably improves multi-round reasoning accuracy, spatial grounding precision, and global–local consistency, establishing a strong baseline for this emerging research direction. Our code is available at RegionReasoner.

## 1 INTRODUCTION

Recent advances in large Vision-Language Models have led to remarkable progress in multimodal reasoning tasks. Leading systems such as OpenAI GPT-4o/GPT-o1 (Hurst et al., 2024; Jaech et al., 2024), Gemini-2.5 (Gemini Team et al., 2023), DeepSeek (DeepSeek-AI et al., 2025; Wu et al., 2024) and VL-Rethinker (Wang et al., 2025a) have achieved state-of-the-art results on benchmarks including MathVista (Lu et al., 2024), MMMU (Yue et al., 2024), and MEGA-Bench (Chen et al., 2025). These methods follow a common paradigm: they first process multimodal inputs, extract textual cues, and then perform chain-of-thought reasoning (Wei et al., 2022) exclusively in the text space. Within the vision community, two particularly relevant lines have pushed the field forward. VisionReasoner (Liu et al., 2025b) showed that structured perception–reasoning with explicit output tags and reward shaping (e.g., format and geometric rewards) yields robust single-turn grounding and interpretable trajectories. SegLLM (Wang et al., 2025b) demonstrated that multi-round interaction is beneficial for challenging referring segmentation, organizing dialogue-style supervision and evaluation across turns.

VisionReasoner (Liu et al., 2025b) establishes a strong single-turn paradigm with structured tags and base rewards (format and geometry). However, when naively stacked into a multi-round protocol, two issues arise: (i) the framework does not require the reasoning to explicitly cite regions grounded in previous turns, so reference propagation across rounds is brittle—credit assignment becomes ambiguous and coordinate hallucinations are hard to detect; and (ii) its reward shaping primarily targets the final outputs (boxes/points) and tag validity, providing little signal to stabilize the reasoning trace itself as dialogue context accumulates, which leads to semantic drift between global descriptions and local evidence at deeper rounds. Conversely, SegLLM (Wang et al., 2025b) brings multi-round interaction into referring segmentation, but it does not model a thinking process: there is no explicit, verifiable reasoning trace to check whether references are truly used, no mechanism to enforce

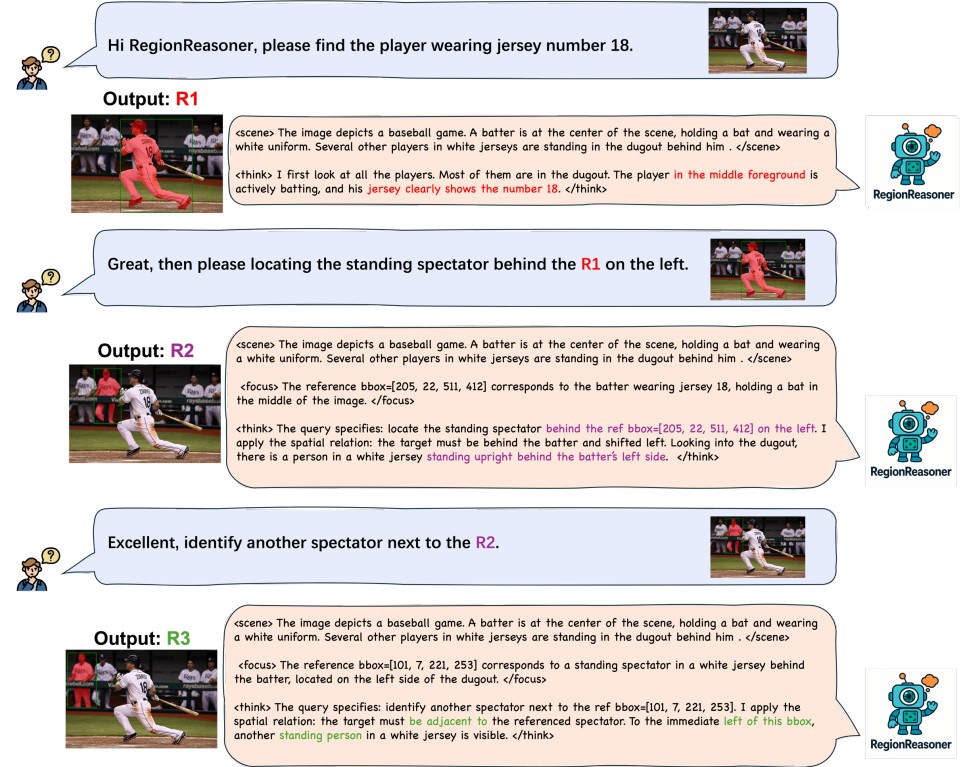

Figure 1: **RegionReasoner in a three–round, region-grounded dialogue.** At round $t$, the user query may refer to a region localized earlier (R1/R2). For each turn, RegionReasoner produces a structured trajectory: `<scene>` (global context), `<focus>` (caption restricted to the referenced region with serialized coordinates, e.g., bbox=[$x_1$, $y_1$, $x_2$, $y_2$]), `<think>` (reasoning that *explicitly cites* the reference and the required spatial relation), and `<answer>` (final localization). The example shows correct citation and stable multi-round grounding for "behind the R1 on the left" and "next to the R2", illustrating how explicit reference use and coherent global–local descriptions support consistent localization as the dialogue deepens.

global–local semantic coherence, and no learning signal to shape intermediate steps; the supervision remains mask-centric and does not naturally extend to detection. These gaps motivate our design in Fig. 1: each round produces a structured trajectory (`<scene>`, `<focus>`, `<think>`, `<answer>`) with reference-grounded thinking and a global–local consistency signal; rewards act on the reasoning trace and the final prediction, enabling interpretable and verifiable multi-round grounding.

Building on these insights, we present **RegionReasoner**, a reinforcement learning-optimized framework that extends VisionReasoner's structured outputs to the multi-round setting studied by SegLLM and directly addresses the limitations above. First, we introduce *reference-grounded thinking*: every reasoning step must explicitly cite the required reference bounding boxes in `<think>`. A dedicated citation reward and a penalty for missing or hallucinated citations make evidence use verifiable and stabilize reference propagation across rounds. Second, we propose a *global–local consistency* reward that aligns keywords from the global scene caption (`<scene>`) and region-level captions (`<focus>`) with the reasoning trace (`<think>`); a lightweight spatial/comparison/localization lexicon further encourages explicit relational language and reduces semantic drift as context accumulates. Third, we assemble RegionDial-Bench, a multi-round benchmark spanning detection and segmentation with per-turn metrics and train/evaluation splits constructed from public referring datasets, enabling quantitative assessment of reasoning accuracy, grounding fidelity, and global–local alignment under iterative interaction. Taken together, these contributions complement VisionReasoner's structured, reward-shaped formulation and SegLLM's multi-round protocol by explicitly modeling and reinforcing the reasoning process across turns.

Our RegionReasoner is trained with reinforcement learning using structured rewards that target grounding fidelity, global–local semantic alignment, and task correctness. On RegionDial-Bench,

RegionReasoner consistently outperforms strong Vision-Language Models and task-specific baselines on both referring segmentation and detection. Two empirical patterns emerge: (i) gains are most pronounced at later turns, reflecting slower error accumulation and more stable reference propagation; and (ii) the signals act complementarily—reference citation chiefly reduces coordinate hallucinations and improves reuse/refinement of prior regions, while global–local consistency stabilizes the semantics of the reasoning trace in scenes with weak spatial cues. Ablations corroborate these trends, with the combined signals delivering the strongest multi-round performance and qualitative trajectories showing verifiable citations and coherent scene–region descriptions across turns.

## 2 RELATED WORK

**Post-training for vision-language models.** Post-training techniques, including instruction tuning and reinforcement learning (RL), have become essential for adapting large Vision-Language Models (VLMs) to complex multimodal reasoning tasks. Early efforts such as LLaVA (Liu et al., 2023), LLaVA-OV (Li et al., 2024), Infinity-MM (Gu et al., 2024), MAmmoTH-VL (Guo et al., 2025) , LISA (Lai et al., 2024), PixelLM (Ren et al., 2024), and GLAMM (Rasheed et al., 2024) demonstrate that scaling instruction-tuning datasets and diversifying task formats can significantly improve generalization across multimodal benchmarks. More recent work, such as VL-Rethinker (Wang et al., 2025a), further explores post-training for reasoning, introducing techniques like selective sample replay to address instability in RL optimization. Unlike these approaches, which mainly focus on single-pass or text-only reasoning, our work enforces explicit spatial grounding and global–local consistency within multi-round visual reasoning.

**Reinforcement learning for multimodal reasoning.** RL has emerged as a powerful tool for enhancing the reasoning and decision-making of VLMs. Vision-R1 (Huang et al., 2025) and Video-R1 (Feng et al., 2025) integrate RL to improve spatial grounding and temporal reasoning, respectively, while VLM-R1 (Shen et al., 2025) applies RL to fine-grained grounding tasks. Pixel Reasoner (Su et al., 2025) further incentivizes pixel-space reasoning with curiosity-driven exploration. Visionary-R1 (Xia et al., 2025) mitigates shortcut behaviors in visual reasoning with explicit RL signals, and the Self-Rewarding VLM (Li et al., 2025) adopts a reasoning-decomposition strategy where the model first generates image captions before deriving answers. Other efforts, such as OpenVLThinker (Deng et al., 2025) and LMM-R1 (Peng et al., 2025), adopt policy optimization methods like PPO (Schulman et al., 2017) to train VLMs as interactive decision-makers. Despite these advances, most RL-based approaches focus on single-pass reasoning or rely on textualized visual inputs, limiting their ability to enforce explicit spatial grounding or multi-step consistency. In contrast, RegionReasoner leverages RL to jointly optimize multi-round reasoning accuracy, region-level grounding fidelity, and global–local semantic alignment, providing a more structured training signal than prior RL-based methods.

**Multi-round visual understanding.** SegLLM (Wang et al., 2025b) explores multi-round interaction for referring segmentation and shows the value of dialogue-style supervision and evaluation, but it does not model explicit reasoning trajectories or incorporate RL signals, making it difficult to verify evidence use or enforce global–local semantic coherence. VisionReasoner (Liu et al., 2025b) provides structured, reward-shaped perception–reasoning in a single-turn setting without reference propagation across rounds. In this context, SegLLM also releases a multi-round segmentation benchmark; our *RegionDial-Bench* complements it by adding explicit reasoning-oriented design and per-turn evaluation for *both* referring detection and referring segmentation, enabling analysis of reasoning accuracy, grounding fidelity, and global–local alignment under iterative interaction.

## 3 PROBLEM FORMULATION WITH REGIONDIAL-BENCH

**Multi-round region-grounded reasoning.** Given an image $I$ and a dialogue of $T$ turns with queries $\{q_t\}_{t=1}^{T}$, a model interacts with the visual scene over multiple turns. Each turn $t$ may include a set of *reference boxes* $\mathcal{B}_t^{\text{ref}} = \{[x_1, y_1, x_2, y_2]\}$ that are propagated from earlier turns or externally provided, specifying regions that subsequent queries should condition on. Let $\mathcal{M}_{t-1}$ denote the dialogue memory up to turn $t-1$ (e.g., previously localized regions or textual context). A policy $\pi_\theta$ produces a turn-level output

$$o_t \sim \pi_\theta\big(\cdot \mid I, q_t, \mathcal{B}_t^{\text{ref}}, \mathcal{M}_{t-1}\big),$$

where $o_t$ instantiates the task-specific prediction at turn $t$ (e.g., a 2D bounding box for detection, a point/mask for segmentation, or a count). The memory is updated as $\mathcal{M}_t = \mathcal{M}_{t-1} \cup \{(q_t, o_t)\}$ to enable *reference propagation* across turns. An episode ends at $T$; evaluation is conducted per turn and aggregated over the dialogue.

**Tasks: detection and segmentation.** We consider two instantiations of $o_t$: (i) *referring detection*, where $o_t$ is a 2D box for the referred region; and (ii) *referring segmentation*, where $o_t$ is a sparse point or mask for the referred region. Later turns may refer to regions predicted earlier via $\mathcal{B}_t^{\text{ref}}$. For detection, we report per-turn AP at IoU$= 0.5$ (AP$_{50}$) and the average across turns. For segmentation, we report per-turn generalized IoU (gIoU) averaged over images and then over turns.

**RegionDial-Benchmark.** To operationalize this setting, we construct a multi-round benchmark, dubbed **RegionDial-Bench** , from the public referring-expression datasets RefCOCO+ and Ref-COCOg. These corpora are built on the MSCOCO image backbone and provide (i) high-quality instance-level bounding boxes and segmentation masks, (ii) human-written referring expressions that are tightly aligned with individual objects, and (iii) multiple expressions per image. This combination makes them particularly well-suited for constructing dialogue-style multi-round grounding tasks without introducing new annotations or relying on synthetic text. In RegionDial-Bench , we consolidate image-wise related expressions into dialogues and rewrite later turns to include explicit references to previously localized boxes. Concretely, our resource contains *RefCOCO+ Multi-turn* (715 images, 2355 turns) and *RefCOCOg Multi-turn* (1,580 images, 4405 turns). Training dialogues are generated by decomposing multi-object instructions and propagating ground-truth references to later turns; test dialogues use model-predicted references, so errors made at early turns can propagate through the dialogue. Construction rules, spatial-relation templates, statistics, and examples are detailed in Appendix B, which also discusses how the same procedure can be extended to other referring-expression datasets with sufficiently dense annotations.

## 4 REGIONREASONER

In this section, we present *RegionReasoner* and its reinforcement learning framework for multi-round visual reasoning. We first formalize the end-to-end pipeline (§4.1), then describe the model architecture and structured I/O design (§4.2). We next detail the reference-grounded and global–local consistency rewards (§4.3), and finally outline the training procedure (§4.4). An overview of the complete framework is provided in Appendix Figure D.

### 4.1 PIPELINE FORMULATION

**Inputs and state.** At turn $t$, the agent observes the image $I$, the current textual query $q_t$, an optional set of reference boxes $\mathcal{B}_t^{\text{ref}} = \{[x_1^{(k)}, y_1^{(k)}, x_2^{(k)}, y_2^{(k)}]\}$ (propagated or newly provided), and a memory $\mathcal{M}_{t-1}$ that stores structured outputs from previous turns. We serialize $\mathcal{B}_t^{\text{ref}}$ and $\mathcal{M}_{t-1}$ into the prompt to make them available to the model.

**Policy and action space.** RegionReasoner is an auto-regressive VLM policy $\pi_\theta$ that generates a *structured text action* composed of four tagged blocks $y_t = (s_t, f_t, h_t, a_t)$ with tags `<scene>`, `<focus>`, `<think>`, `<answer>`. Let $y_t = (w_{t,1}, \ldots, w_{t,N_t})$ denote the token sequence for the whole action; then

$$\pi_\theta(y_t \mid I, q_t, \mathcal{B}_t^{\text{ref}}, \mathcal{M}_{t-1}) = \prod_{n=1}^{N_t} \pi_\theta(w_{t,n} \mid I, q_t, \mathcal{B}_t^{\text{ref}}, \mathcal{M}_{t-1}, w_{t,<n}). \tag{1}$$

Constrained decoding enforces the tag schema and JSON validity for `<answer>`, while allowing free-form natural language in `<scene>`, `<focus>`, and `<think>`.

**Turn update and termination.** After decoding finishes (upon emitting the end token or the closing `</answer>` tag), we parse $a_t$ to obtain task outputs (e.g., 2D boxes or points) and update the memory:

$$\mathcal{M}_t = \mathcal{M}_{t-1} \cup \{(s_t, f_t, h_t, a_t)\}. \tag{2}$$

A multi-round episode consists of $T$ turns (fixed or query-driven). The per-turn reward $R(t)$ is computed from $(s_t, f_t, h_t, a_t)$ and aggregated across turns (Sec. 4.3, 4.4).

**Compact notation for the loop.** For brevity, we denote the one-turn transition produced by the policy as

$$(s_t, f_t, h_t, a_t) \sim \pi_\theta(\cdot \,|\, I, q_t, \mathcal{B}_t^{\text{ref}}, \mathcal{M}_{t-1}), \qquad \mathcal{M}_t \leftarrow \mathcal{M}_{t-1} \cup \{(s_t, f_t, h_t, a_t)\}. \qquad (3)$$

## 4.2 REGIONREASONER MODEL

**Unified perception–reasoning backbone.** RegionReasoner extends the unified perception–reasoning framework of VisionReasoner (Liu et al., 2025b) to a multi-round setting, where each turn emits a structured and verifiable trajectory. The model is initialized from a large VLM backbone and performs chain-of-thought reasoning purely in text, while remaining *explicitly* grounded to image regions through serialized bounding-box references. Each turn-$t$ output is organized into four tagged blocks: a global scene caption $s_t$ (`<scene>`), a localized caption $f_t$ tied to a provided reference box (`<focus>`, optional), a reasoning trace $h_t$ (`<think>`), and a JSON answer $a_t$ (`<answer>`). Constrained decoding with schema and tag guards ensures format validity, supports automatic post-hoc parsing, and prevents untagged content from leaking into `<answer>`.

**Reference-grounded thinking.** To improve verifiability and reduce free-form hallucination, RegionReasoner requires that *reasoning must cite evidence*. When a query specifies references, the prompt encodes the set $\mathcal{B}_t^{\text{ref}} = \{[x_1^{(k)}, y_1^{(k)}, x_2^{(k)}, y_2^{(k)}]\}$ in a canonical textual form and instructs the model to reason with *verbatim* coordinate mentions inside `<think>`. The same coordinates are injected in $q_t$ so attention aligns with the intended regions across turns. During decoding, $h_t$ must explicitly reference the used boxes and, when relevant, name spatial relations (e.g., "to the right of bbox $[x_1, y_1, x_2, y_2]$"). This design yields a causal chain from evidence to conclusion that is parsable into cited coordinates $\mathcal{S}(h_t)$ and directly comparable to $\mathcal{B}_t^{\text{ref}}$, enabling automatic grounding checks and precise credit assignment in RL. In multi-round interaction, previously cited boxes can be re-used or refined; the explicit citation acts as a stable interface across turns, which improves temporal coherence of the reasoning trajectory and curbs region drift.

**Global–local semantic consistency.** Iterative reasoning often breaks down when global descriptions and local evidence diverge; to prevent this, RegionReasoner jointly produces $s_t$ (global) and $f_t$ (localized to the reference) before generating $h_t$, and then enforces that the semantics of $s_t$ and $f_t$ are reflected within $h_t$. Concretely, a lightweight deterministic pipeline extracts keyword sets $\mathcal{K}(s_t)$, $\mathcal{K}(f_t)$, and $\mathcal{K}(h_t)$ (lowercasing, stop-word removal, lemmatization, and a noun/object filter). We later compute asymmetric overlaps $\text{Ov}(s_t, h_t)$ and $\text{Ov}(f_t, h_t)$ as part of the reward (Sec. 4.3), pushing the model to propagate entities and relations from the global and local captions into the reasoning itself. Making `<think>` the alignment nexus—rather than correcting only at the final answer—yields finer-grained RL signals, better consistency across turns, and improved spatial reasoning, especially when $h_t$ is encouraged to include localization lexicon (e.g., *left/right/inside/overlap/next to*) together with explicit box mentions.

**Task output without extra heads.** Detection and segmentation are expressed directly through the JSON `<answer>` without introducing task-specific heads. For segmentation, we use sparse `point_2d` outputs to probe masks following our benchmark protocol; evaluation employs IoU/Dice or point-based matching as appropriate. This head-free design keeps the learning signal unified: structural validity and geometric precision are attributed to `<answer>`, while grounding fidelity and global–local agreement are attributed to `<think>` in conjunction with `<scene>` and `<focus>`. The result is a closed loop where interpretable trajectories, verifiable references, and final predictions are optimized jointly under multi-round supervision.

## 4.3 REWARD FUNCTIONS

We optimize RegionReasoner with reinforcement learning, shaping both intermediate reasoning and final predictions. Besides the base rewards inherited from prior work (Liu et al., 2025b), *Thinking Format*, *Answer Format*, *Non-Repeat*, *Bboxes IoU*, *Bboxes L1*, and *Points L1*, we introduce two multi-round objectives that explicitly encode (i) citation of required references inside the reasoning trace and (ii) semantic alignment between global and local evidence.

**Notation.** At turn $t$, the model outputs $s_t$ (`<scene>`), $f_t$ (`<focus>` if any), $h_t$ (`<think>`), and $a_t$ (`<answer>`). Required references are $\mathcal{B}_t^{\text{ref}} = \{b_k^{\text{ref}}\}$ (possibly empty). A lightweight extractor $\mathcal{K}(\cdot)$

returns keyword sets (lowercasing, stop-word removal, lemmatization, noun/object filter). We parse bbox mentions from $h_t$ as $\mathcal{S}(h_t)$ and use $\text{kw}(h_t) \in \{0, 1\}$ to flag bbox-related tokens.

**Reference citation reward.** To make the reasoning verifiable and grounded, the trace must explicitly cite the referenced boxes when they are required. We reward correct citation and penalize hallucinated coordinates:

$$R_{\text{ref}}(t) = \begin{cases} 1, & \mathcal{B}_t^{\text{ref}} = \varnothing, \\ \lambda\,\text{kw}(h_t) \;+\; \mu\,\dfrac{|\mathcal{S}(h_t) \cap \mathcal{B}_t^{\text{ref}}|}{\max\left(|\mathcal{S}(h_t)|, 1\right)}, & \text{otherwise,} \end{cases} \qquad R_{\text{ref}}(t) \leftarrow \begin{cases} \eta\,R_{\text{ref}}(t), & \mathcal{S}(h_t) \setminus \mathcal{B}_t^{\text{ref}} \neq \varnothing, \\ R_{\text{ref}}(t), & \text{otherwise,} \end{cases}$$

(4)

with $\lambda = \mu = 1.0$, $\eta = 0.5$, and clipping $R_{\text{ref}}(t) \in [0, 2]$.

**Global–local consistency reward.** To keep the reasoning coherent with both global scene context and localized evidence, we align $h_t$ with $s_t$ and (when present) $f_t$. Let the asymmetric keyword overlap be

$$\text{Ov}(X, Y) = \frac{|\mathcal{K}(X) \cap \mathcal{K}(Y)|}{\max\left(|\mathcal{K}(X)|, 1\right)}. \tag{5}$$

We also include a light logic prior $\ell(h_t) \in [0, 1]$ counting spatial/comparison/localization terms (capped at 1). The consistency reward is

$$R_{\text{cons}}(t) = w_s\,\text{Ov}(s_t, h_t) + w_f\,\not\Vdash\!\left[\mathcal{B}_t^{\text{ref}} \neq \varnothing\right]\text{Ov}(f_t, h_t) + w_\ell\,\ell(h_t), \tag{6}$$

with $w_s = 1.0$, $w_f = 0.6$, $w_\ell = 0.4$, clipped to $[0, 2]$.

**Total per-turn objective and episode return.** Let $R_{\text{base}}(t)$ denote the base rewards from (Liu et al., 2025b) (*Thinking/Answer Format*, *Non-Repeat*, *Bboxes IoU/L1*, *Points L1*). The per-turn reward aggregates as

$$R(t) = R_{\text{base}}(t) + \alpha\,R_{\text{ref}}(t) + \beta\,R_{\text{cons}}(t), \tag{7}$$

where $\alpha = \beta = 1$ by default. Each component is normalized to $[0, 2]$ prior to aggregation to balance scales, and the episode return is $\sum_t R(t)$ over turns. Compared to baselines, these rewards are used only as internal training signals; all evaluation metrics remain purely geometry-based (AP and gIoU) and are computed identically for all models.

## 4.4 TRAINING

We optimize the policy $\pi_\theta$ with GRPO (Shao et al., 2024) over multi-turn rollouts. For each batch, the model generates structured actions $y_t = (s_t, f_t, h_t, a_t)$ at turns $t = 1 \ldots T$ conditioned on $(I, q_t, \mathcal{B}_t^{\text{ref}}, \mathcal{M}_{t-1})$ as defined in Sec. 4.1. Per-turn rewards follow the decomposition in Sec. 4.3—$R_{\text{base}}, R_{\text{ref}}, R_{\text{cons}}$—with componentwise normalization to $[0, 2]$; the episode return is $\sum_{t=1}^{T} R(t)$.

**Objective.** We optimize the clipped policy objective GRPO (Shao et al., 2024) on the autoregressive likelihood of the structured action (cf. equation 1):

$$\mathcal{L}_{\text{clip}}(\theta) = \mathbb{E}\Big[\min\Big(\rho_t(\theta)\,\hat{A}_t,\ \text{clip}\big(\rho_t(\theta), 1 - \epsilon, 1 + \epsilon\big)\,\hat{A}_t\Big)\Big], \quad \rho_t(\theta) = \frac{\pi_\theta(y_t \mid I, q_t, \mathcal{B}_t^{\text{ref}}, \mathcal{M}_{t-1})}{\pi_{\theta_{\text{old}}}(y_t \mid I, q_t, \mathcal{B}_t^{\text{ref}}, \mathcal{M}_{t-1})}.$$

**Advantage estimation and value targets.** Let $s_t = (I, q_t, \mathcal{B}_t^{\text{ref}}, \mathcal{M}_{t-1})$ denote the turn-$t$ state and $r_t$ the per-turn reward. We use a learned value head $V_\phi(s)$ and compute advantages with GAE:

$$\delta_t = r_t + \gamma\,V_\phi(s_{t+1}) - V_\phi(s_t), \qquad \hat{A}_t = \sum_{l=0}^{T-t} (\gamma\lambda)^l\,\delta_{t+l}.$$

Each dialogue is a finite episode; the last turn $T$ is terminal, so we set

$$V_\phi(s_{T+1}) = 0.$$

The value target is $\hat{R}_t = \hat{A}_t + V_\phi(s_t)$ and the critic is trained with $\mathcal{L}_{\text{value}} = \frac{1}{2}\left(V_\phi(s_t) - \hat{R}_t\right)^2$. We add a small entropy bonus to encourage exploration and, optionally a KL penalty to a frozen reference policy for stability:

$$\mathcal{L}_{\text{total}} = \mathcal{L}_{\text{clip}} + c_v\,\mathcal{L}_{\text{value}} - c_e\,\mathbb{H}[\pi_\theta(\cdot|s_t)] + \beta\,\text{KL}(\pi_\theta(\cdot|s_t) \,\|\, \pi_{\text{ref}}(\cdot|s_t)).$$

Table 1: **Detection on RegionDial-Bench with 7-round dialogues.** Columns report per-round AP (R1–R7) and the mean across turns for RefCOCO+ Multi-turn and RefCOCOg Multi-turn. RegionReasoner-7B achieves the top averages on both splits and maintains larger margins at later rounds, reflecting stronger robustness to error accumulation.

| Method | RefCOCO+ Multi-turn (AP ↑) | | | | | | | | RefCOCOg Multi-turn (AP ↑) | | | | | | | |
|---|---|---|---|---|---|---|---|---|---|---|---|---|---|---|---|---|
| | R1 | R2 | R3 | R4 | R5 | R6 | R7 | Avg | R1 | R2 | R3 | R4 | R5 | R6 | R7 | Avg |
| Qwen2−VL−7B | 6.2 | 8.5 | 6.5 | 5.4 | 7.5 | 3.6 | 3.5 | 6.7 | 7.8 | 6.2 | 3.5 | 3.5 | 5.6 | 4.0 | 5.0 | 6.1 |
| Qwen2.5−VL−7B | 65.5 | 49.0 | 48.1 | 36.5 | 30.0 | 38.2 | 25.9 | 49.9 | 63.9 | 43.7 | 39.0 | 37.9 | 42.2 | 43.2 | 33.8 | 49.7 |
| Seg−Zero−7B | **90.5** | 71.2 | 73.6 | 59.6 | 48.8 | 58.2 | 48.2 | 73.1 | 85.3 | 61.8 | 61.6 | 64.8 | 70.0 | 69.6 | 68.8 | 71.1 |
| VisionReasoner−7B | 88.3 | 74.7 | 75.8 | 64.2 | 56.3 | 57.3 | 47.0 | 74.8 | 85.0 | 65.8 | 66.8 | **69.3** | 68.3 | 75.2 | 72.5 | 73.6 |
| **RegionReasoner −7B** | 89.3 | **83.2** | **81.6** | **69.6** | **61.9** | **69.1** | **64.7** | **80.7** | **87.1** | **73.7** | **71.8** | 68.6 | **75.0** | **78.4** | **75.0** | **78.2** |

A sliding memory $\mathcal{M}_{t-1}$ preserves prior turns under context budget, and a light turn-depth curriculum gradually increases the maximum $T$ early in training. Constrained decoding enforces tag/schema and JSON validity so that rewards are well-defined both for intermediate reasoning (`<scene>`/`<focus>`/`<think>`) and final outputs (`<answer>`). Compared to SegLLM (Wang et al., 2025b), which performs multi-round segmentation without explicit reasoning traces or RL, our training aligns interpretable, reference-grounded thinking with global–local consistency under a unified multi-round objective.

## 5 EXPERIMENTS

### 5.1 EXPERIMENTAL SETTINGS

**Benchmark and protocol.** We evaluate under the multi-round setting in Sec. 3 on *RegionDial-Bench* (RefCOCO+ / RefCOCOg Multi-turn). Detailed descriptions of the dataset construction procedure, together with quantitative statistics, are provided in Appendix B. In addition, following the evaluation protocol of VisionReasoner (Liu et al., 2025b), we also report results under the single-round setting.

**Base model.** RegionReasoner-7B is initialized from Qwen2.5-VL-7B (Bai et al., 2025) (7B parameters). We keep the vision–language backbone intact and optimize it end-to-end with reinforcement learning; no additional task-specific heads are introduced.

**Implementation details.** RegionReasoner-7B is trained with GRPO (Shao et al., 2024) using the rewards in Sec. 4.3. Constrained decoding enforces tag/schema validity and JSON correctness. We use the backbone's vision tokenizer and input resolution; the maximum turn depth $T$ matches the dialogue length. Training uses a global batch size of 16 with $K=8$ rollout samples per prompt (per step). The initial learning rate is $1\times10^{-6}$ with weight decay 0.01. All experiments run on $4\times$ NVIDIA H100 GPUs; total training time is about 10 hours. Unless noted, we fix random seeds and use identical multi-turn contexts and references across methods; shared evaluation scripts ensure consistent aggregation.

**Baselines.** We compare RegionReasoner-7B with strong VLMs and task-specialized models: Qwen2.5-VL-7B (Bai et al., 2025) and Qwen2-VL-7B (Wang et al., 2024); Seg-Zero-7B (Liu et al., 2025a) (segmentation-centric); VisionReasoner-7B (Liu et al., 2025b) (structured perception–reasoning in a single-turn setting); and SegLLM (Wang et al., 2025b) (multi-round segmentation without explicit thinking or RL). All methods are evaluated under the same multi-turn protocol with reference propagation; for models without structured reasoning, we adapt prompts to accept referenced boxes.

### 5.2 MAIN RESULTS

**Referring detection under multi-round interaction.** Table 1 reports AP on RegionDial-Bench. RegionReasoner-7B attains the highest turn-average on both splits, improving over VisionReasoner-7B by 5.9 points on RefCOCO+ (80.7 vs. 74.8) and 4.6 points on RefCOCOg (78.2 vs. 73.6). Against Seg-Zero-7B, the gains are 7.6 (RefCOCO+) and 7.1 (RefCOCOg) points. Late-turn improvements are pronounced: on RefCOCO+ the margins at R5/R6/R7 are +5.6/+11.8/+17.7 over VisionReasoner-7B. These results indicate that explicit reference citation and global–local consistency preserve localization quality as dialogue context deepens.

Table 2: **Segmentation on RegionDial-Bench with 7-round dialogues.** Columns report per-round gIoU (R1–R7) and the mean across turns for RefCOCO+ Multi-turn and RefCOCOg Multi-turn. RegionReasoner-7B attains the highest averages on both splits and sustains larger gains at later rounds, indicating stronger robustness to error accumulation in multi-round settings.

| Method | RefCOCO+ Multi-turn (gIoU ↑) | | | | | | | | RefCOCOg Multi-turn (gIoU ↑) | | | | | | | |
|---|---|---|---|---|---|---|---|---|---|---|---|---|---|---|---|---|
| | R1 | R2 | R3 | R4 | R5 | R6 | R7 | Avg | R1 | R2 | R3 | R4 | R5 | R6 | R7 | Avg |
| Qwen2−VL−7B | 12.2 | 9.0 | 6.5 | 5.3 | 6.3 | 6.3 | 9.5 | 9.4 | 8.5 | 11.6 | 8.8 | 10.0 | 7.0 | 6.4 | 4.4 | 9.3 |
| Qwen2.5−VL−7B | 56.5 | 43.3 | 41.4 | 34.5 | 23.4 | 33.6 | 24.9 | 43.6 | 53.8 | 36.3 | 35.5 | 31.6 | 37.3 | 36.8 | 28.6 | 42.1 |
| Seg−Zero−7B | **78.6** | 62.8 | 64.0 | 51.6 | 42.4 | 50.8 | 46.7 | 64.0 | 72.3 | 52.3 | 53.5 | 55.4 | 59.4 | 59.5 | 58.3 | 60.5 |
| SegLLM−7B | 71.1 | 71.7 | 70.4 | 58.7 | 41.9 | 39.2 | 30.3 | 60.7 | 68.9 | 55.3 | 50.5 | 47.7 | 47.3 | 37.8 | 25.4 | 56.7 |
| VisionReasoner−7B | 75.6 | 65.0 | 65.9 | 54.9 | 46.6 | 48.9 | 40.8 | 64.3 | 69.5 | 52.7 | 55.4 | 56.0 | 57.8 | 64.1 | 57.6 | 59.9 |
| **RegionReasoner −7B** | 76.4 | **73.1** | **72.0** | **58.8** | **51.3** | **59.4** | **54.6** | **69.6** | **73.9** | **62.9** | **60.7** | **58.9** | **64.4** | **66.8** | **63.3** | **66.5** |

Table 3: **Ablation on RegionReasoner components for detection**. Left: components toggled. Right: *Single-Round* vs. *Multi-Round*. Base rewards follow Liu et al. (2025b). "Ref-cite" enforces explicit bbox citation in <think>; "Consist." is the keyword-overlap consistency reward; "Logic" is the lightweight spatial/comparison/localization prior. Ref-cite and Consist. both help, their combination yields additional gains, and the full model provides the strongest multi-round AP.

| Components | Toggles | | | Single-Round | | Multi-Round | |
|---|---|---|---|---|---|---|---|
| | Ref-cite | Consist. | Logic | RefCOCO+ | RefCOCOg | RefCOCO+ | RefCOCOg |
| Base only (no new signals) | ✗ | ✗ | ✗ | 87.9 | 87.5 | 74.8 | 73.6 |
| + Ref-cite only | ✓ | ✗ | ✗ | **88.6** | **88.4** | 78.9 | 77.1 |
| + Ref-cite + Consist. | ✓ | ✓ | ✗ | 88.1 | 88.2 | 80.2 | 77.6 |
| **+ Ref-cite + Consist. + Logic** | ✓ | ✓ | ✓ | 87.7 | 87.9 | **80.7** | **78.2** |

Table 4: **Ablation on RegionReasoner components for segmentation**. Same toggles as Table 3. Overall, either Ref-cite or Consist. improves over the base, their combination brings further gains, and the full model attains the best multi-round performance.

| Components | Toggles | | | Single-Round | | Multi-Round | |
|---|---|---|---|---|---|---|---|
| | Ref-cite | Consist. | Logic | RefCOCO+ | RefCOCOg | RefCOCO+ | RefCOCOg |
| Base only (no new signals) | ✗ | ✗ | ✗ | 74.9 | 71.3 | 64.3 | 59.9 |
| + Ref-cite only | ✓ | ✗ | ✗ | **76.9** | **74.4** | 67.9 | 63.6 |
| + Ref-cite + Consist. | ✓ | ✓ | ✗ | 74.0 | 70.9 | 68.3 | 65.8 |
| **+ Ref-cite + Consist. + Logic** | ✓ | ✓ | ✓ | 74.1 | 71.2 | **69.6** | **66.5** |

**Referring segmentation under multi-round interaction.** Table 2 summarizes gIoU on RegionDial-Bench. RegionReasoner -7B attains the highest turn-average on both RefCOCO+ and RefCOCOg and exceeds all baselines across most rounds. Relative to VisionReasoner-7B, the average gains are 5.3 points on RefCOCO+ and 6.6 points on RefCOCOg; RegionReasoner also improves over SegLLM by about 8.9 and 9.8 points on RefCOCO+ and RefCOCOg, respectively. The gap widens at deeper turns (R7), indicating that explicit reference citation together with global–local consistency mitigates error accumulation and preserves spatial fidelity as dialogue context grows. Representative trajectories are shown in Fig. 2, where RegionReasoner explicitly cites referenced boxes in <think>, maintains agreement between scene- and region-level descriptions, and resists nearby distractors, while VisionReasoner tends to drift at later turns.

## 5.3 ABLATION ANALYSIS

We study the contribution of each signal using Tables 3 and 4, which report single- and multi-round results on RefCOCO+ and RefCOCOg.

**Effect of reference citation (Ref-cite).** Enforcing explicit citation of the referenced box in <think> consistently boosts multi-round performance for both tasks, with the largest gains at later turns where error carryover is strongest. Citation turns cross-turn dependence into verifiable evidence use: the policy learns to reuse or refine previously grounded coordinates, which curbs drift and avoids spurious boxes. In the single-round protocol, a nontrivial subset of queries still provides a reference region (from our spatial-relation templates), so $R_{\text{ref}}$ is active and yields measurable improvements by tying the reasoning trace to the given coordinates and aligning <think> with <answer>; when no reference is provided, this term is neutral. By contrast, the consistency and logic signals chiefly stabilize semantics and relational language across turns, hence their effects are most visible in the multi-round setting.

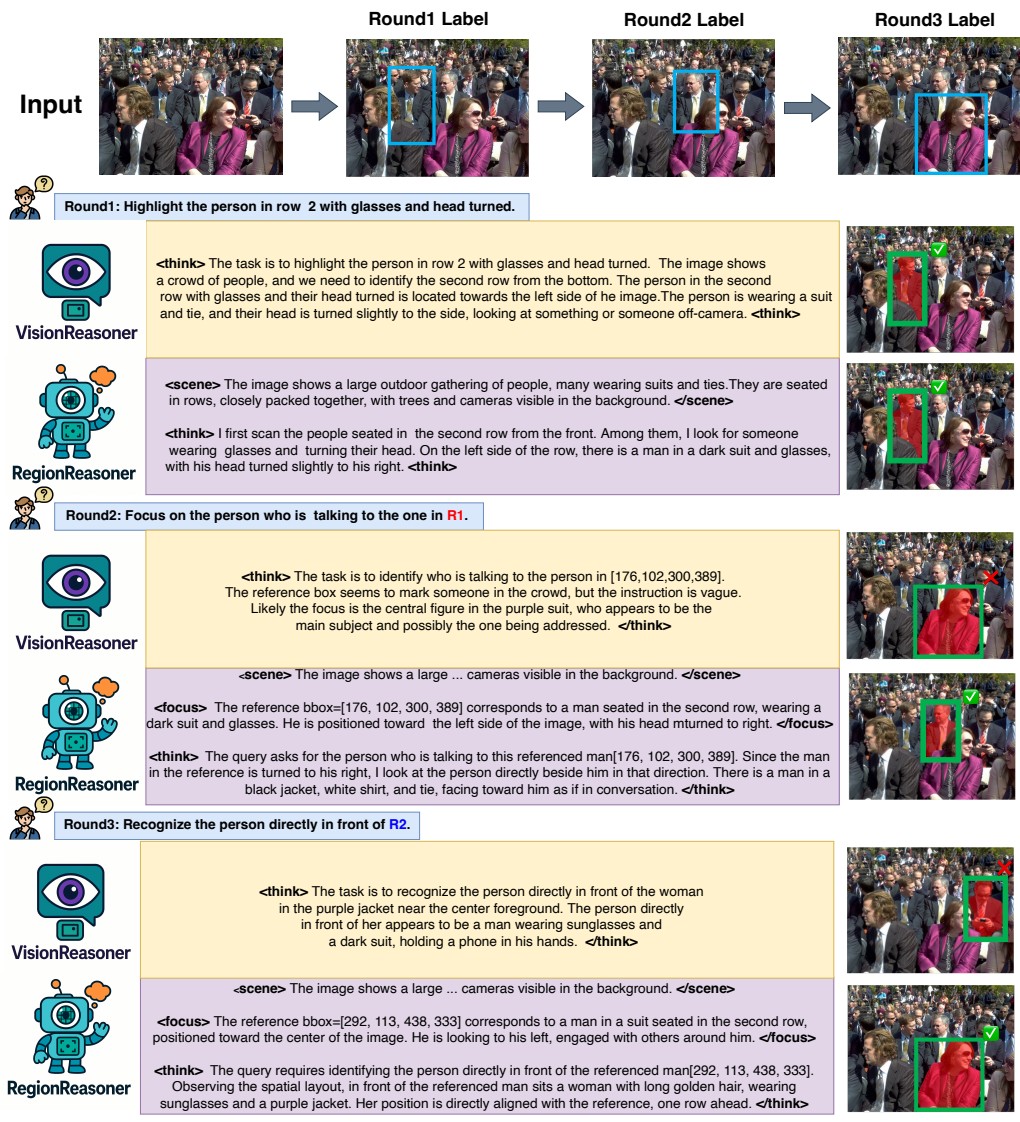

Figure 2: **Qualitative multi-round trajectories (R1–R3) on our RegionDial-Bench .** Each panel shows RegionReasoner vs. VisionReasoner. Blue boxes mark the ground-truth reference regions for each round. Green boxes denote predicted detection boxes, while red masks denote predicted segmentation outputs. Checkmarks and crosses indicate prediction correctness. RegionReasoner consistently *cites* the reference coordinates inside `<think>` and aligns its reasoning with global (`<scene>`) and local (`<focus>`) descriptions, yielding stable localization in later rounds. VisionReasoner, lacking explicit citation, is prone to semantic drift or neighbor confusion when context accumulates.

**Effect of global–local consistency (Consist.).** Aligning keywords between global scene descriptions and localized region captions strengthens the reasoning trace, with particularly clear benefits on RefCOCO+ where spatial hints in the query are weak. The key effect is semantic anchoring: nouns and objects echoed in `<think>` keep the trajectory focused on the same entities across turns, which limits off-topic attention and stabilizes segmentation quality in cluttered scenes.

**Effect of the logic prior.** Adding the lightweight spatial/comparison/localization lexicon yields small yet persistent gains, most visible at deeper turns. Encouraging phrases such as *inside, next to, left of* increase reward density for partially correct reasoning and nudges the model to articulate relations explicitly. This makes the trace easier to verify and helps the policy recover when two candidates are visually similar.

**Depth robustness and single- vs. multi-round difficulty.** Across datasets and tasks, single-round results (Round 1) are consistently higher than their multi-round counterparts, which reflects an intrinsic difficulty gap rather than an artifact of a particular model. In the single-round setting, the policy only needs to resolve one query against the image. In contrast, later rounds must both interpret the current query and correctly reuse and propagate previously predicted boxes as references. Any localization error at an early turn is carried forward and compounds over subsequent turns, so the effective difficulty increases with turn depth. All compared methods exhibit this depth-dependent degradation in Tables 3 and 4, highlighting multi-turn error accumulation and robust reference propagation as central challenges for grounded dialogue. The full RegionReasoner configuration degrades more slowly with turn index than any variant without citation or without consistency: its trajectories remain parseable and self-consistent, which limits the accumulation of small localization errors over long dialogues. For all ablations, we keep schema and JSON checks enabled to isolate learning effects from parsing noise.

## 6 CONCLUSIONS

We introduced multi-round visual reasoning and presented **RegionReasoner**, a reinforcement-learning framework that couples interpretable, reference-grounded thinking with global–local semantic alignment. The model emits structured trajectories, and is optimized with two targeted rewards: a reference–citation signal that enforces explicit grounding to cited boxes and a consistency signal that aligns global and region-level captions with the reasoning trace. To enable systematic evaluation, we released **RegionDial-Bench**, multi-turn training and testing resources spanning detection and segmentation. Experiments on RefCOCO+ and RefCOCOg under multi-round protocols show consistent improvements, especially at deeper turns where cascading errors typically degrade performance.

**Ethics statement.** This work proposes RegionReasoner and RegionDial-Bench for multi-round visual reasoning. We do not collect new human data or elicit sensitive attributes. All images and annotations used to build **RegionDial-Bench** are derived from *public* referring datasets (RefCOCO+, RefCOCOg) under their licenses; our multi-turn dialogues are programmatic reformulations of existing annotations, with no additional human labeling. We do not attempt to infer demographics, identities, or other sensitive information. Potential misuse includes applying the method to private imagery without consent or deploying it in settings that require privacy guarantees; we discourage such uses and recommend adherence to data-governance policies and applicable licenses.

**Reproducibility statement.** All compared models (e.g., Qwen2.5-VL-7B, Seg-Zero-7B, VisionReasoner-7B, SegLLM) and datasets are publicly accessible. Methodology, reward design, and training procedure are detailed in Sections 4 and 4.4; benchmark construction, evaluation protocols, and baselines are in Section 5. To facilitate replication, we will release code, **RegionDial-Bench** conversion scripts, prompts, reward configurations, and evaluation scripts upon acceptance. Compute details: RegionReasoner-7B is trained with policy-gradient RL on $4\times$ NVIDIA H100 GPUs for approximately 10 hours; batch size, optimizer settings, and other hyperparameters are reported in Section 5. We will provide random seeds and exact checkpoints to ensure reproducibility.

## ACKNOWLEDGMENTS

This work was supported by the European Union's Horizon Europe research and innovation programme under grant agreement number 101214398 (ELLIOT).

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

## A    LLM USAGE STATEMENT

We used a large language model (ChatGPT) solely for grammar checking and language polishing of the manuscript text. It did not contribute to research ideation, method design, experiments, data analysis, or result generation; all technical content was authored and verified by the authors.

## B    MULTI-ROUND BENCHMARKS

**Training set construction.** We extend the ~7k single-turn samples from VisionReasoner (Liu et al., 2025b) into ~10k dialogue samples. The expansion comes from decomposing multi-object instructions into sequential sub-queries, such that a single original sample may yield multiple turns. Later rounds are explicitly grounded to the bounding boxes predicted in earlier rounds, while single-object queries remain in single-turn form without references.

For example, the instruction *"a black and white dog laying down, looking away from the camera" and "standing dog"* is reformulated into: (1) "a black and white dog laying down, looking away from the camera"; (2) "find the standing dog, next to bbox=[0,457,374,672]". Here, the coordinates [0,457,374,672] denote the ground-truth bounding box of the "a black and white dog laying down" from Round 1, injected into Round 2 as a *reference bounding box*. An illustration of this reformulation process is shown in Figure 3. This process increases the total number of training samples to about 10k, though not all samples involve reference propagation.

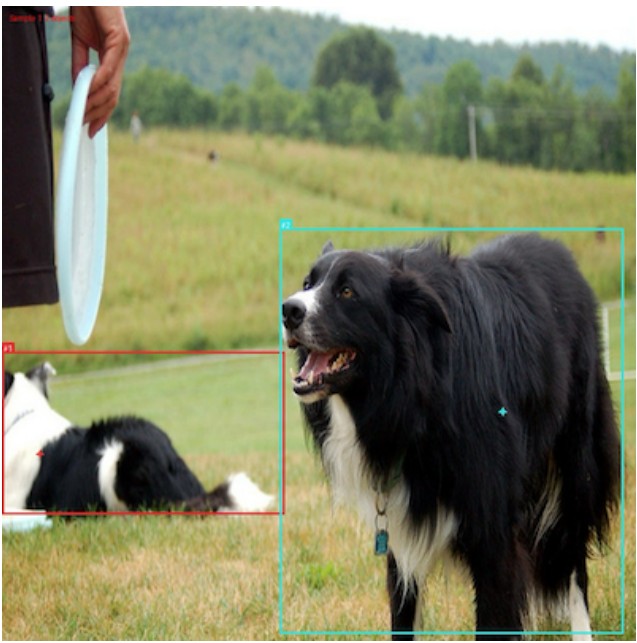

Figure 3: Example of training data construction. Round 1 localizes the "laying dog" (red box). Round 2 reformulates the query into "standing dog, next to bbox=[0,457,374,672]" (blue box).

To diversify spatial interactions, we introduce eight spatial relation templates covering adjacency, directional, containment, and overlap/contact relations (Table 5).

**Test set construction.** RegionDial-Bench is constructed entirely from the public referring expression benchmarks RefCOCO+ and RefCOCOg, using only their official test splits. We reuse the original images, human-written referring expressions, and ground-truth bounding boxes/masks without introducing any new images or annotations. In the original datasets, each test sample is a single-turn example consisting of one query and one target region, but many such samples share the same underlying image.

Table 5: Eight spatial relation templates used to construct multi-round dialogues. They cover four categories of spatial interactions: adjacency (next to), directional (above, below, left, right), containment (inside), and contact/overlap (overlapping with, touching).

| Relation Type | Template |
|---|---|
| Adjacency | `next to bbox=[x1,y1,x2,y2]` |
| Directional (above) | `above bbox=[x1,y1,x2,y2]` |
| Directional (below) | `below bbox=[x1,y1,x2,y2]` |
| Directional (left) | `to the left of bbox=[x1,y1,x2,y2]` |
| Directional (right) | `to the right of bbox=[x1,y1,x2,y2]` |
| Containment | `inside bbox=[x1,y1,x2,y2]` |
| Overlap | `overlapping with bbox=[x1,y1,x2,y2]` |
| Touching | `touching bbox=[x1,y1,x2,y2]` |

We first group all RefCOCO+/g test samples by image and then merge the queries associated with the same image into coherent multi-round dialogues. As illustrated in Figure 4, Round 1 localizes the "man in blue shirt" (red box) with ground-truth box [47,107,303,466]. For each subsequent round, we deterministically inject the bounding box predicted at an earlier round (or the ground-truth box during training) into the query as an explicit reference token (e.g., "bbox=[47,107,303,466]"), while keeping the original target labels unchanged. This procedure yields two multi-turn evaluation sets: RefCOCO+ Multi-turn (715 images, 2355 dialogue turns) and RefCOCOg Multi-turn (1,580 images, 4405 dialogue turns), with dialogue lengths ranging from 1 to 7 rounds. Table 6 reports the per-round sample counts and resulting dialogue-length distribution. Object categories strictly follow those in the original RefCOCO+/g datasets (COCO-style categories for RefCOCO+, with testA dominated by the "person" class, and 78 categories for RefCOCOg).

**Dataset choice.** Our goal is to study multi-round referring grounding with both detection and segmentation, under a protocol that requires: (i) high-quality instance-level masks and bounding boxes, (ii) human-written referring expressions aligned with specific objects, and (iii) multiple expressions per image to support dialogue-style construction. RefCOCO+ and RefCOCOg jointly satisfy all these requirements. Both datasets are built on the MSCOCO dataset (Lin et al., 2014), and therefore inherit its large-scale instance segmentation and detection annotations with well-established train/val/test splits. Crucially, they are explicitly designed for referring-expression grounding, offering clean natural-language queries that correspond to individual object instances. Furthermore, many images contain several distinct referring expressions, which is essential for forming coherent multi-round dialogues over the same scene.

Using raw MSCOCO alone would require generating or mining referring expressions as a preprocessing step, introducing an additional modeling component orthogonal to our focus on multi-round grounding. Visual Genome (Krishna et al., 2017) provides rich relational annotations and region descriptions, but its instance segmentation masks are sparse and less consistent, making the link between text and fine-grained segmentation less reliable. For our setting—where each turn requires an accurate region mask or bounding box as a reference—this mismatch becomes a serious limitation.

Within the RefCOCO family, we choose RefCOCO+ and RefCOCOg rather than including RefCOCO itself. Although they share the same underlying MSCOCO images, the linguistic design differs: RefCOCO+ forbids location words, yielding appearance-centric expressions, while RefCOCOg contains longer and more descriptive queries covering 78 categories. Using RefCOCO+ and RefCOCOg thus provides a diverse combination of concise and rich expressions without introducing near-duplicate supervision from RefCOCO, whose differences stem primarily from annotation rules rather than visual content.

We refer to these resources collectively as **RegionDial-Bench**, the first manually curated multi-round benchmark for reference-grounded reasoning. Unlike prior multi-round resources constructed via GPT-style automatic rewriting, RegionDial-Bench is built from human-authored referring expressions combined with deterministic reference propagation from ground-truth boxes, avoiding LLM-induced artifacts and yielding more reliable evaluation.

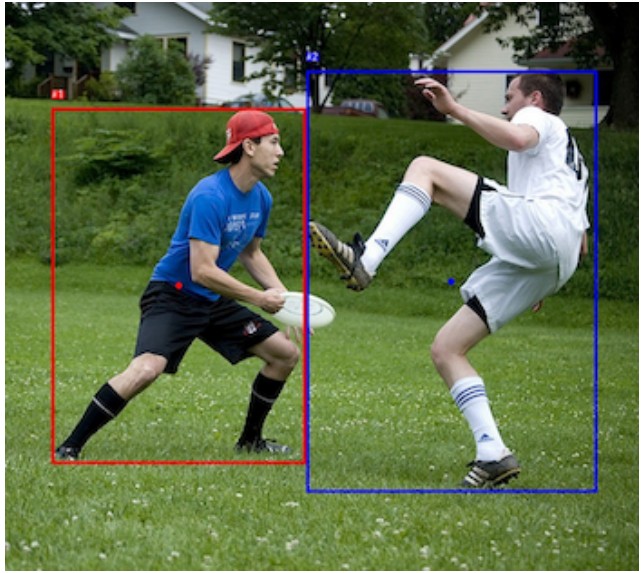

Figure 4: Example from RefCOCO+ Multi-turn illustrating the construction pipeline in RegionDial-Bench. Round 1 localizes the "man in blue shirt" (red box) with ground-truth box [47,107,303,466]. This box is then injected into Round 2 as an explicit reference, reformulating the query into "Who is next to bbox=[47,107,303,466]?" to localize the "man in white shirt" (blue box).

Table 6: Per-round dialog-turn statistics for RegionDial-Bench. Dialogue lengths range from 1 to 7 rounds; the bottom row reports the total number of dialogue turns in each multi-turn test set.

| Round | RefCOCO+ Multi-turn (dialog turns) | RefCOCOg Multi-turn (dialog turns) |
|-------|------------------------------------|------------------------------------|
| 1 | 715 | 1,580 |
| 2 | 715 | 1,580 |
| 3 | 310 | 570 |
| 4 | 260 | 290 |
| 5 | 160 | 180 |
| 6 | 110 | 125 |
| 7 | 85 | 80 |
| **Total** | **2,355** | **4,405** |

## C    INSTRUCTION SCHEMA

To guide the policy model toward producing structured reasoning trajectories, we design a unified *instruction schema* for training in Table 7. At inference time, we use a unified *instruction schema* in Table 8, which is shared by all baseline methods to ensure fair comparison. This schema specifies how user queries, reference bounding boxes, and reasoning steps are serialized into a consistent prompt format, inspired by prior approaches (Liu et al., 2025b; Wang et al., 2025b).

## D    REGIONREASONER FRAMEWORK

Figure 5 illustrates the overall framework of **RegionReasoner**. The model is built upon the Qwen2.5-VL-7B backbone and is optimized with two reinforcement learning objectives: the *reference citation reward*, which enforces explicit grounding to previously localized objects, and the *global–local consistency reward*, which aligns holistic scene understanding with reference-based reasoning. This framework summarizes how user instructions, reference propagation, and reward shaping are integrated to enable coherent multi-round reasoning.

Table 7: Instruction schema used during **training**.



**Training Instruction Schema**

`<image>`

**Task:** "Please find "{Question}" with bboxs and points."

**Reference guidance:** If a reference bbox is provided (e.g., `above/ below/ to the left of/ to the right of/ inside/ overlapping with/ touching bbox=[x1,y1,x2,y2]`), use it as spatial guidance.

**Steps:** 1) In `<scene> </scene>`, give a concise global scene description.
2) In `<focus> </focus>`, describe what is visible inside the reference bbox (if provided). (do not output the final answer or target label here).
3) In `<think> </think>`, reason over the whole image by combining the global scene and the reference bbox relation. Explicitly state which spatial relation from the question you apply (e.g., "target is above the reference"), and use it to constrain the search over the scene to locate the target object(s). If multiple candidates exist, compare them and pick the closest match.
4) In `<answer> </answer>`, output the bbox(es) and point(s) for the target object(s) in JSON.

**Format:** `<scene>` global description of the image here `</scene>`
`<focus>` description of reference bbox content here `</focus>`
`<think>` thinking process here `</think>`
`<answer>{Answer}</answer>`



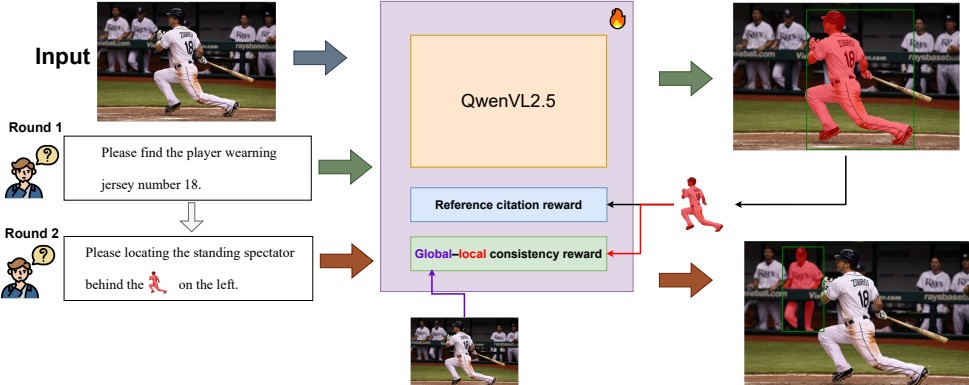

Figure 5: Framework of **RegionReasoner**. The model processes multi-round queries with Qwen2.5-VL-7B, guided by two complementary reward signals: (1) the *reference citation reward*, ensuring explicit grounding to previously predicted objects, and (2) the *global–local consistency reward*, enforcing alignment between holistic and reference-based reasoning.

## E ADDITIONAL QUALITATIVE RESULTS

To complement the quantitative results in the main paper, we provide additional qualitative visualizations in Figure 6. These examples illustrate how our model performs multi-round reference-grounded reasoning on challenging cases from **RegionDial-Bench**. In particular, they highlight the model's ability to propagate references across dialogue turns and maintain consistent localization. Beyond

Table 8: Instruction schema used during **inference**.

---

**Inference Instruction Schema**

`<image>`

**Task:** "Please find "{Question}" with bboxs and points."

**Reference guidance:** If a reference bbox is provided (e.g., `above/ below/ to the left of/ to the right of/ inside/ overlapping with/ touching bbox=[x1,y1,x2,y2]`), use it as spatial guidance only.

**Steps:** 1) In `<scene> </scene>`, give a concise global scene description.
2) If a reference bbox exists, in `<focus> </focus>` describe ONLY what is visible inside that bbox (do not output the final answer or target label here).
3) In `<think> </think>`, reason over the whole image by combining the global scene and the reference bbox relation. Explicitly state which spatial relation from the question you apply (e.g., "target is above the reference"), and use it to constrain the search over the scene to locate the target object(s). If multiple candidates exist, compare them and pick the closest match.
4) In `<answer> </answer>`, output the bbox(es) and point(s) for the target object(s) in JSON.

**Format:** `<scene>` global scene description `</scene>`
`<focus>` description of reference bbox content (if provided bbox=[x1,y1,x2,y2]) `</focus>`
`<think>` reasoning that applies the spatial relation to the scene and narrows to the final target(s) `</think>`
`<answer>`{Answer}`</answer>`

---

the three-turn examples shown above, we also include cases with longer dialogue chains. Figure 7 illustrates a four-turn dialogue from **RegionDial-Bench**, demonstrating how our model propagates references across multiple levels of reasoning.

## F    GENERALIZATION TO EXTERNAL BENCHMARK

To assess whether RegionReasoner generalizes beyond RegionDial-Bench, we further evaluate the model on the $V^*$ benchmark (Wu & Xie, 2024), which explicitly targets attribute-level and spatial visual search in multimodal LLMs. We follow the official $V^*$ evaluation protocol and compare RegionReasoner-7B with GPT-4V, SEAL (Wu & Xie, 2024) (the method proposed in $V^*$), Qwen2.5-VL-7B, and VisionReasoner-7B. The quantitative results are shown in Table 9. SEAL achieves the highest overall score because it incorporates an explicit visual-search mechanism specifically engineered for the $V^*$ benchmark and tightly coupled to the LLaVA architecture, making it incompatible with the Qwen2.5-VL family without substantial re-engineering. Within the Qwen2.5-VL family, RegionReasoner attains the strongest overall performance among all models *without* a dedicated visual-search module. RegionReasoner demonstrates particularly large gains on the Spatial dimension (+7.9 over Qwen2.5-VL and +7.9 over VisionReasoner), indicating that reference-grounded reasoning and global–local consistency rewards improve spatial localization and visual search in a way that transfers beyond our proposed benchmark. Note that RegionReasoner is trained exclusively on RegionDial-Bench and never on $V^*$, further confirming the generalizability of our approach.

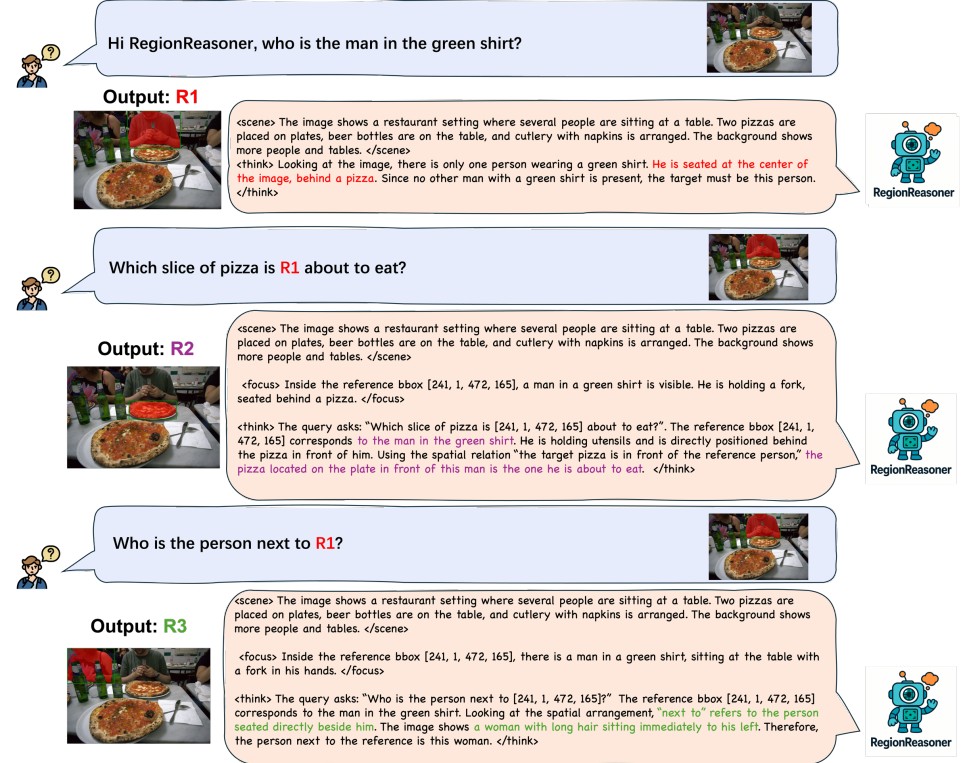

Figure 6: Multi-round qualitative example from **RegionDial-Bench**. The dialogue contains three rounds: (1) "Who is the man in the green shirt?" → localized as the bounding box [241,1,472,165]. (2) "Which slice of pizza is R1 about to eat?" → where R1 refers to the bounding box predicted in Round 1, and the model localizes the corresponding pizza slice. (3) "Who is the person next to R1?" → again using the bounding box from Round 1 as a reference, the model identifies the adjacent person.

Table 9: Evaluation on the V* benchmark. RegionReasoner achieves the best performance among models based on the Qwen2.5-VL backbone and shows strong generalization to attribute-level and spatial visual search without using a specialized visual-search module.

| Model | Attribute ↑ | Spatial ↑ | Overall ↑ | Visual Search Mechanism |
|---|---|---|---|---|
| GPT-4V | 51.30 | 60.52 | 54.97 | no |
| SEAL | 74.78 | 76.31 | 75.39 | yes |
| Qwen2.5-VL-7B | 72.17 | 60.52 | 67.53 | no |
| VisionReasoner-7B | 75.62 | 60.52 | 69.63 | no |
| **RegionReasoner-7B** | **75.65** | **68.42** | **72.77** | no |

# G STANDARD SINGLE-ROUND REC AND RES RESULTS

We report standard single-round referring expression comprehension (REC; detection) and referring expression segmentation (RES) results on the RefCOCO+ and RefCOCOg benchmarks. In this conventional setting, each referring expression is evaluated independently, without any multi-round dependencies. As shown in Table 10, the model achieves strong performance on both REC and RES in the standard single-round setting across RefCOCO+ and RefCOCOg. These results demonstrate that the model maintains solid grounding capability under the conventional single-turn protocol.

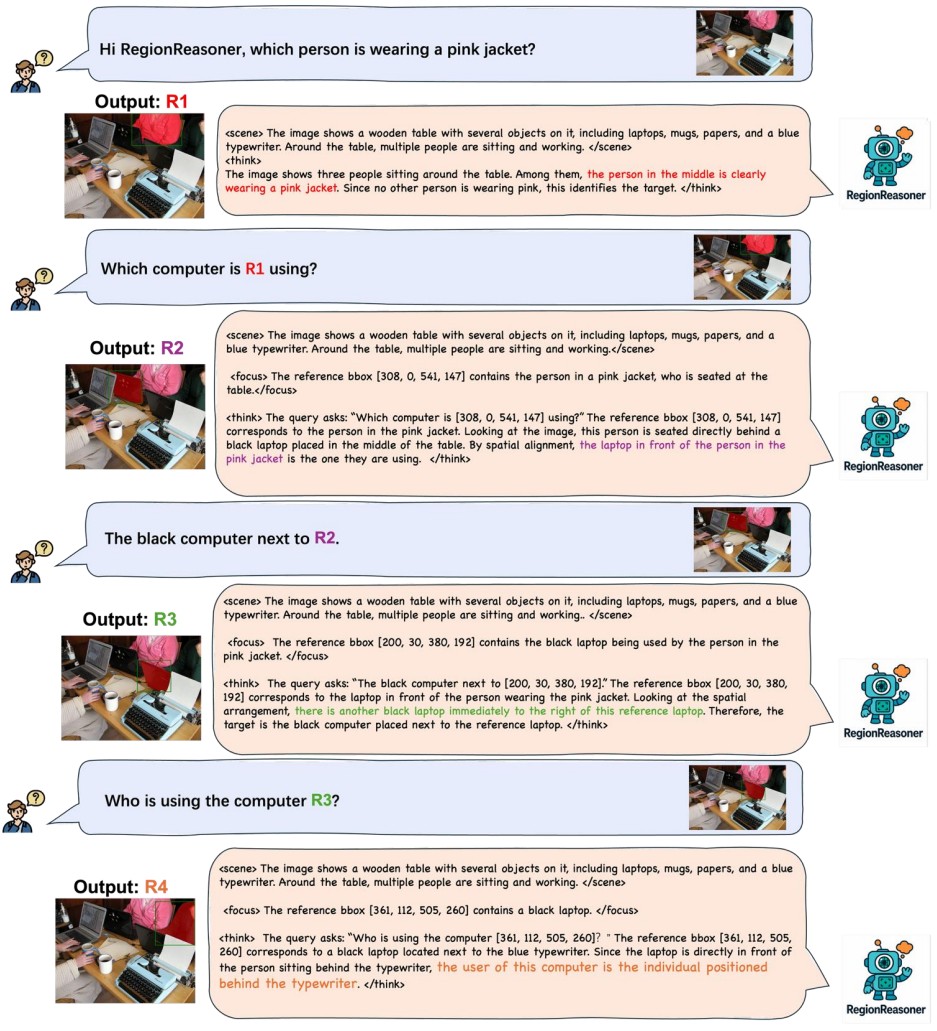

Figure 7: Four-turn qualitative example from **RegionDial-Bench**. The dialogue proceeds as follows: (1) "Which person is wearing a pink jacket?" → localized as bounding box R1. (2) "Which computer is R1 using?" → model grounds the computer associated with R1, denoted as bounding box R2. (3) "The black computer next to R2." → model localizes the black computer adjacent to R2, denoted as bounding box R3. (4) "Who is using the computer R3?" → finally, the model grounds the user of the black computer R3.

Table 10: Standard single-round REC (detection AP) and RES (segmentation gIoU) on RefCOCO+ and RefCOCOg test sets.

| Model | Seg. RefCOCO+ | Seg. RefCOCOg | Det. RefCOCO+ | Det. RefCOCOg |
|---|---|---|---|---|
| Qwen2-VL-7B | 65.7 | 63.5 | 76.5 | 78.2 |
| Qwen2.5-VL-7B | 76.8 | 72.8 | 88.2 | 85.7 |
| VisionReasoner-7B | 74.9 | 71.3 | 87.9 | 87.5 |
| **RegionReasoner-7B** | **76.9** | **74.4** | **88.6** | **88.4** |

## H SENSITIVITY STUDY OF REWARD WEIGHTS $\alpha$ AND $\beta$

To examine the sensitivity of the per-turn reward

$$R(t) = R_{\text{base}}(t) + \alpha\, R_{\text{ref}}(t) + \beta\, R_{\text{cons}}(t),$$

Table 11: Sensitivity of RegionReasoner to variations in reward weights $\alpha$ and $\beta$. Metrics are averaged over multi-turn detection (Det) and segmentation (Seg) on the RefCOCO+ and RefCOCOg benchmarks.

| $\alpha$ / $\beta$ Setting | RefCOCO+ Det | RefCOCOg Det | RefCOCO+ Seg | RefCOCOg Seg |
|---|---|---|---|---|
| $\alpha = 1.0,\ \beta = 0.5$ | 79.7 | 77.6 | 68.1 | 65.7 |
| $\alpha = 0.5,\ \beta = 1.0$ | 79.9 | 77.7 | 68.4 | 65.9 |
| $\alpha = 1.5,\ \beta = 1.0$ | 80.4 | 78.1 | 69.2 | 66.2 |
| $\alpha = 1.0,\ \beta = 1.5$ | 80.2 | 78.0 | 68.9 | 66.0 |
| $\alpha = 1.0,\ \beta = 1.0$ (**default**) | **80.7** | **78.2** | **69.6** | **66.5** |

we conduct a small-scale study varying the coefficients $\alpha$ and $\beta$ around the default setting used throughout the main paper ($\alpha = \beta = 1.0$). All reward components are normalized to the range $[0, 2]$, so setting both coefficients to 1.0 provides a balanced weighting between reference-citation fidelity and global–local semantic consistency.

Table 11 reports performance when either coefficient is halved or increased by 50% while holding the other fixed. Across all four metrics—detection and segmentation on RefCOCO+ and RefCOCOg—the overall trends remain stable. Increasing $\alpha$ slightly improves robustness at deeper turns by strengthening reference grounding, while increasing $\beta$ slightly improves performance in scenes with weaker spatial cues. The balanced setting $\alpha = \beta = 1.0$ offers the best trade-off across datasets and metrics, without requiring dataset-specific tuning. The results indicate that RegionReasoner is robust to moderate changes in reward weighting, and the default balanced configuration is an effective choice across all benchmarks.

# I    LIMITATIONS

Our consistency reward relies on lightweight keyword extraction and a hand-crafted logic prior, which may miss paraphrases or subtle relations. Grounding is enforced via boxes and points rather than full masks, and our constrained schema may introduce sensitivity to formatting. Extending RegionReasoner to richer relation graphs, mask-level grounding, longer dialogues and videos, and learnable entailment-based consistency is a promising direction. In the meantime, we hope RegionDial-Bench and RegionReasoner establish a strong baseline that spurs further research on interpretable, reference-grounded multi-round visual reasoning.

