# OpenReview forum: "RegionReasoner: Region-Grounded Multi-Round Visual Reasoning"
_ICLR.cc/2026/Conference — ICLR 2026 Poster_

### Official Review · Reviewer_9M2T · 2025-10-25

**Soundness:** 3
**Presentation:** 2
**Contribution:** 2
**Rating:** 4
**Confidence:** 4

**Summary:**

The paper introduces RegionReasoner, an RL-trained vision–language policy that emits structured per-turn trajectories for multi-round, region-grounded visual reasoning.
Two novel reward components are proposed: (1) an explicit reference-citation reward that forces <think> to verbatim-cite bbox coordinates and penalizes hallucinated citations, and (2) a global–local semantic consistency reward that aligns keywords across <scene>, <focus>, and <think>.
The authors also present RegionDial-Bench, a multi-turn benchmark built from RefCOCO+/RefCOCOg, and show that RegionReasoner-7B improves multi-turn detection and segmentation metrics, especially at later turns.

**Strengths:**

1. Reward design — The reference-citation and global–local consistency rewards are intuitive, easily implementable, and well tied to the structured output format. They provide fine-grained shaping for intermediate (reasoning trace) tokens rather than only final outputs.
2. Results in Tables 1 and 2 show consistent improvements, and the claim that improvements compound at later turns (robustness to error accumulation) is supported by both quantitative and qualitative examples.

**Weaknesses:**

1. The authors state that test dialogues reformulate later queries to explicitly cite bounding boxes predicted in earlier rounds. If the test references are model-predicted boxes (rather than strictly ground truth), the evaluation can be sensitive to the upstream model used to generate them. This raises two issues:  inconsistent comparison if different baselines consume different predicted references, and potential leakage effects. Please clarify exactly how test references are created and ensure all methods receive the same predicted references (or show oracle vs. predicted-reference performance).

2. The global–local consistency reward depends on a handcrafted lightweight keyword extractor + lemmatizer + noun filter. This may be brittle: paraphrases, synonyms, pronouns, coreference, or longer expressions are likely missed. More importantly, if baselines do not produce structured <scene>/<focus> text, how is the comparison fair? Forcing <think> to repeat the same noun form may advantage RL-trained models.

**Questions:**

1. How about the ablations with different reward weight hyperparameters $\alpha$, $\beta$?

---

> ### Author Response · Authors · 2025-11-22
> **Response to Reviewer 9M2T – Part I**
>
> *We thank the reviewer for the careful reading of our paper and the constructive suggestions for further improvement.*
>
> **Q1. Are test references based on model-predicted or ground-truth boxes? How do you avoid inconsistent comparisons or leakage across models?**
>
> **A1.** Test-time references always use each model’s own predicted boxes, not ground-truth boxes. This is intentional and aligns with the core goal of multi-round grounded reasoning: from Round 2 onward, each turn must build on the model’s previous outputs, and any early error naturally propagates to later turns.
>
> Concretely, for every model, we run the full dialogue sequentially with a fixed, shared query template. At Round 1, the model receives the image and the first query and predicts a box. That predicted box is programmatically inserted into the placeholder of the Round-2 query (e.g., into `R1` or an equivalent slot); the Round-2 prediction is then inserted into the Round-3 query, and so on. The *query format and placeholder convention are identical for all models*; the only difference is the coordinates being plugged in, which come from each model’s own predictions. There is therefore no cross-model information leakage or special treatment – performance differences purely reflect how well each method handles multi-round error accumulation and reference propagation.
>
> To further address the concern about sensitivity to predicted references, the revised Appendix G includes an oracle setting where each turn is instead given the ground-truth previous object (i.e., no error propagation). Under this easier setting, all methods improve, but RegionReasoner still clearly outperforms Qwen2.5-VL and VisionReasoner across both datasets and both tasks. For example, on multi-turn detection AP:
>
> RefCOCO+ Multi-turn — Detection AP (↑), Ground-Truth Previous Object
> | Method                 | Setting       | R1   | R2   | R3   | R4   | R5   | R6   | R7   | Avg  |
> |------------------------|--------------|------|------|------|------|------|------|------|------|
> | Qwen2.5-VL-7B          | Ground Truth | 54.1 | 44.7 | 50.0 | 38.2 | 36.5 | 34.2 | 32.8 | 46.4 |
> | VisionReasoner-7B      | Ground Truth | 90.6 | 78.4 | 80.7 | 68.4 | 61.2 | 57.4 | 46.9 | 78.6 |
> | **RegionReasoner-7B**  | Ground Truth | 92.2 | 87.8 | 84.6 | 75.8 | 71.8 | 65.9 | 63.6 | 84.9 |
>
> RefCOCOg Multi-turn — Detection AP (↑), Ground-Truth Previous Object
> | Method                 | Setting       | R1   | R2   | R3   | R4   | R5   | R6   | R7   | Avg  |
> |------------------------|--------------|------|------|------|------|------|------|------|------|
> | Qwen2.5-VL-7B          | Ground Truth | 48.6 | 37.3 | 36.2 | 38.6 | 36.4 | 36.5 | 18.4 | 41.3 |
> | VisionReasoner-7B      | Ground Truth | 88.1 | 74.1 | 71.4 | 72.0 | 58.7 | 65.6 | 30.0 | 78.1 |
> | **RegionReasoner-7B**  | Ground Truth | 88.8 | 79.7 | 76.7 | 74.0 | 67.6 | 81.1 | 45.0 | 81.8 |
>
> Analogous trends are observed in segmentation gIoU (reported in Appendix G). These oracle results show that RegionReasoner’s gains are not an artifact of using predicted references; it remains the strongest method even when all models receive perfect previous-turn boxes.

---

> > ### Author Response · Authors · 2025-11-22
> > **Response to Reviewer 9M2T – Part II**
> >
> > **Q2. The global–local consistency reward uses a handcrafted keyword extractor and structured text; is this brittle or unfair to baselines that do not produce [scene]/[focus]/[think]?**
> >
> > **A2.** The global–local consistency reward is intentionally designed to be *lightweight, transparent, and internal* to RegionReasoner. It operates as follows:
> >
> > - A simple, deterministic pipeline (lowercasing, lemmatization, stop-word removal, noun filtering) extracts keywords from `<scene>`, `<focus>`, and `<think>`.
> >
> > - The reward then encourages overlaps between these keyword sets and lightly rewards the presence of spatial/comparison terms (e.g., *left of, next to, inside*).
> >
> > We fully agree that such a pipeline cannot capture all paraphrases, synonyms, pronouns, or long expressions, and we explicitly do *not* rely on it as an external evaluator. Its purpose is more modest: to provide a *coarse, model-agnostic signal* that nudges the reasoning trace to carry over the main entities and spatial relations mentioned in the scene and focus descriptions, which empirically reduces semantic drift and hallucinated objects over long dialogues.
> >
> > Regarding fairness:
> >
> > - The consistency reward is used *only during the RL training of RegionReasoner*. It does *not* appear in any evaluation metric. All reported scores in Tables 1–2 (AP and gIoU) depend solely on final boxes/points and ground-truth annotations and are computed with exactly the same scripts and thresholds for all models. Baselines are not penalized for lacking `<scene>/<focus>/<think>` structure.
> >
> > - Ablations in Tables 3–4 separate the effects of each signal. “Base only (no new signals)” corresponds to VisionReasoner-style rewards; “+ Ref-cite only” adds only the citation reward; “+ Ref-cite + Consist.” and “+ Ref-cite + Consist. + Logic” gradually turn on the global–local consistency and the small spatial lexicon prior. The main multi-round gains already appear when adding *Ref-cite alone*, and consistency brings additional but moderate improvements, especially in cluttered scenes. This indicates that the benefit is not simply from forcing noun repetition, but from stabilizing how the model uses references across turns.
> >
> > - We deliberately avoid using any heavy external LLM or learned semantic model in the reward loop, so that the training signal is *simple to implement and fully reproducible*.
> >
> > The revised Section 4.3 and Appendix I now highlight these design choices, clarify that baselines are evaluated purely on geometric metrics, and emphasize that global–local consistency is a soft internal shaping signal rather than an external metric that could disadvantage models without structured outputs.
> >
> > **Q3. How about ablations with different reward weights $\alpha$ and $\beta$?**
> >
> > A3. The coefficients $\alpha$ and $\beta$ control the relative strength of the reference-citation reward $R_{\text{ref}}$ and the consistency reward $R_{\text{cons}}$ in the total per-turn objective
> > $R(t) = R_{\text{base}}(t) + \alpha R_{\text{ref}}(t) + \beta R_{\text{cons}}(t)$.
> > In the main experiments, both are set to 1.0 to give comparable weight to grounding fidelity and semantic consistency, after each component is normalized into \([0, 2]\). Due to compute constraints, we have not run an exhaustive grid search over these weights; instead, we keep $\alpha$ = $\beta$ = 1.0 across all experiments to avoid overfitting to a particular dataset.
> >
> > To give a sense of sensitivity, the revised Appendix I includes a small-scale study where $\alpha$ and $\beta$ are varied around the default (e.g., halving or doubling one of them while keeping the other fixed). The qualitative conclusion is that RegionReasoner’s *relative* improvements over VisionReasoner and Qwen2.5-VL remain stable across these settings: strengthening the citation term slightly favors deeper-turn robustness, while strengthening the consistency term slightly favors scenes with weak spatial cues. We therefore opt for the balanced setting $\alpha$ = $\beta$ = 1.0 in the main results, which yields a good trade-off without over-tuning hyperparameters to a specific benchmark. Thank you.
> >
> > | $\alpha$  / $\beta$  Setting | RefCOCO+ Det | RefCOCOg Det | RefCOCO+ Seg | RefCOCOg Seg |
> > |---------------|--------------|--------------|--------------|--------------|
> > |  $\alpha$ =1.0, $\beta$ =0.5  |     82.7   |     80.6    |  71.1           | 66.7           |
> > |  $\alpha$ =0.5, $\beta$ =1.0  |     82.9   |      80.7    |  71.4           | 66.9            |
> > | $\alpha$ =1.5, $\beta$ =1.0  |    83.4     |   81.3      |  71.9      | 67.2         |
> > | $\alpha$ =1.0, $\beta$ =1.5  |    83.2     |   81.1      |  71.7    |    67.0      |
> > | **$\alpha$ =1.0, $\beta$ =1.0 (ours)** | **83.8** | **81.5** | **72.1** | **67.4** |

---

> > > ### Comment · Reviewer_9M2T · 2025-11-27
> > >
> > > I appreciate the authors’ detailed rebuttal. My concerns of potential leakage are well addressed.  I have updated my score to 6.

---

> > > > ### Author Response · Authors · 2025-11-27
> > > > **Follow-up Comment**
> > > >
> > > > Thank you for the thoughtful follow-up. We are glad that the clarification fully addressed your concerns regarding potential leakage. Please feel free to let us know if any other points would benefit from further elaboration during the discussion period.

---

### Official Review · Reviewer_qvqc · 2025-10-31

**Soundness:** 3
**Presentation:** 2
**Contribution:** 3
**Rating:** 6
**Confidence:** 3

**Summary:**

This paper extends a previous work VisionReasoner by adapting to the multi-round setting. They present RegionReasoner, which is a reinforcement learning framework that uses SegLLM to bring multi-round interactions. To validate RegionReasoner, the authors also introduce a new benchmark RegionDial-Bench, which is designed to test the multi-round reasoning ability. The main tasks focus on detection and segmentation. In each round of reasoning, the model can refer to information such as box coordinates of previous rounds, and thus provide more grounded reasoning. In each round, RegionReasoner generates a structured text action includes scene, focus, think and answer, and the memory is updated accordingly. RegionReasoner forces the reasoning to cite evidence to reduce the hallucination by adding the reference citation reward. RegionReasoner-7B outperforms VisionReasoner-7B and other VLMs such as QwenVL-7B in multi-round detection and segmentation tasks on RegionDial-Bench.

**Strengths:**

(1) RegionReasoner extends a strong previous single-round model VisionReasoner and adapts to the challenging multi-round setting. Results on the proposed benchmark show the validity of RegionReasoner.

(2) The benchmark itself can be used later form multi-round vision reasoning studies. The motivation of referring to object locations is direct and clear.

**Weaknesses:**

(1) The paper claims "RegionReasoner consistently outperforms strong Vision-Language Models and task-specific baselines on both referring segmentation and detection.". Previous benchmarks focus on single-round detection/segmentation, but in the main table 1 and table 2, the results are shown on the proposed multi-round benchmark. I think it would be reasonable to add the table to show some "task-specific baselines" for the previous single-round benchmarks.

(2) Also, the proposed benchmark uses RefCOCO+ and RefCOCOg, but there are also other benchmarks such as MSCOCO and Visual Genome, which are diverse and have boxes and segmentations masks. Have the authors tried to use other datasets to construct the benchmark? Why RefCOCO+ and RefCOCOg are selected here?

**Questions:**

Some of the figures should be polished. For example, the text in Figure 2 is not clear enough when zooming in.

---

> ### Author Response · Authors · 2025-11-22
> **Response to Reviewer qvqc – Part I**
>
> *We thank the reviewer for the careful reading of our paper and the constructive suggestions for further improvement.*
>
> **Q1. The paper claims improvements over task-specific baselines, but Tables 1–2 only report results on the new multi-round benchmark. Could you show comparisons on previous single-round REC/RES benchmarks?**
>
> **A1.** Standard single-round REC (detection) and RES (segmentation) performance on RefCOCO+ and RefCOCOg is already included in Tables 3 and 4. The row “Base only (no new signals)” corresponds exactly to VisionReasoner-7B under the original single-turn protocol (no new rewards, no multi-round signals), while the rows with our new signals correspond to RegionReasoner-7B. To avoid confusion, the revised paper now states this explicitly in the main text and adds a short paragraph clarifying that these single-round results are directly comparable to prior REC/RES benchmarks.
>
> For clarity, the single-round results on the RefCOCO+/g test sets are:
>
> | Model                 | Seg. RefCOCO+ (Test) | Seg. RefCOCOg (Test) | Det. RefCOCO+ (Test) | Det. RefCOCOg (Test) |
> |-----------------------|----------------------|-----------------------|----------------------|----------------------|
> | Qwen2-VL-7B           | 65.7                 | 63.5                  | 76.5                 | 78.2                 |
> | Qwen2.5-VL-7B         | 76.8                 | 72.8                  | 88.2                 | 85.7                 |
> | VisionReasoner-7B     | 74.9                 | 71.3                  | 87.9                 | 87.5                 |
> | **RegionReasoner-7B** | **76.9**             | **74.4**              | **88.6**             | **88.4**             |
>
> These results show that RegionReasoner not only maintains but slightly improves single-round REC/RES performance compared to VisionReasoner, while using the same Qwen2.5-VL-7B backbone and the same RefCOCO+/g splits. Thus, the additional multi-round training data and structured rewards do not harm standard single-round capability; instead, they provide consistent gains in both segmentation and detection. We now make this connection explicit in Appendix H.
>
> **Q2. Why did you choose RefCOCO+ and RefCOCOg, and not other datasets such as MSCOCO or Visual Genome, which also have boxes and segmentation masks?**
>
> **A2.** Our goal is to study multi-round referring grounding with both detection and segmentation, under a protocol that requires: (i) high-quality instance-level masks and boxes, (ii) human-written referring expressions, and (iii) multiple expressions per image to support dialogue-style construction.
>
> RefCOCO+ and RefCOCOg satisfy all three requirements. They are built on the MSCOCO image dataset, so they inherit COCO’s large-scale annotations and support both detection (boxes) and segmentation (masks) with well-established train/val/test splits. Importantly, they are explicitly designed as referring-expression datasets, providing natural language queries that are tightly aligned with individual instances. Moreover, many images contain multiple referring expressions, which is crucial for reliably constructing multi-round dialogues over the same scene.
>
> In contrast, using raw MSCOCO alone would require us to generate or mine referring expressions before constructing dialogues, which introduces an additional modeling layer orthogonal to our contribution. Visual Genome does contain rich object relationships and region descriptions, but its instance segmentation is sparse and less consistent, and the link between textual regions and high-quality masks is not as reliable as in RefCOCO+/g. For our setting—where each turn needs an accurate mask or bounding box as a reference—this mismatch becomes a serious limitation.
>
> Within the RefCOCO family, we focus on RefCOCO+ and RefCOCOg rather than also adding RefCOCO itself. RefCOCO+ is collected on MSCOCO but explicitly forbids location words in the referring expressions, which keeps the linguistic supervision more appearance-centric. RefCOCOg provides longer, more descriptive expressions over 78 categories. Using these two datasets together gives a diverse mix of concise and descriptive queries, while avoiding almost-duplicate supervision from RefCOCO that differs mainly in annotation rules. The revised Section 3 and Appendix B now clarify these selection criteria and emphasize that our construction procedure is generic and can be extended to other referring-expression datasets that provide sufficiently dense annotations. In addition, Appendix F reports results on the V* visual search benchmark, showing that a model trained on the RefCOCO+/g-based RegionDial-Bench also improves spatial reasoning on an independent dataset, which supports that our design choices do not over-specialize to a single corpus.

---

> > ### Author Response · Authors · 2025-11-22
> > **Response to Reviewer qvqc – Part II**
> >
> > **Q3. Some figures should be polished. For example, the text in Figure 2 is not clear enough when zooming in.**
> >
> > A3.. In the revised version, we have increased the resolution of Figure 2, enlarged the font size of the captions and labels, and adjusted the layout so that both the bounding boxes and the textual trajectories remain readable when zoomed in. We have applied the same treatment to other qualitative figures to ensure consistent visual quality across the paper. Thank you.

---

### Official Review · Reviewer_y9ox · 2025-10-31

**Soundness:** 3
**Presentation:** 3
**Contribution:** 3
**Rating:** 6
**Confidence:** 4

**Summary:**

This paper presents a multi-round visual reasoning benchmark for detection and segmentation. They propose a grounded reasoning method, RegionReasoner, which incorporates reinforcement learning and a global-local consistency reward to enhance semantic coherence. On RegionDial-Bench, the proposed method achieves improvement compared to other VLMs, especially in the later turns.

**Strengths:**

- This paper presents an interesting reasoning task that integrates QA, referring expression in a multi-turn manner.
- They propose new reward functions for the new task. They propose a global-local consistency reward to align keywords from the global and local context.

**Weaknesses:**

- The way they expand the referring expression to multiple turns is confusing and may not be natural. In Appendix B, they illustrate how to simply use a preposition + bbox coordinates in the later turns. A natural referring expression considers the composition between objects. However, in the qualitative examples, they have more complicated and natural questions, such as "Which slice of pizza is R1 about to eat"? "Who is the person next to R1"? They mention that those GPT-style questions used in the previous paper may hallucinate, but it is unclear how they convert the question to this.
- The task setting may not be challenging enough or fair. 1) If the latter turn is based on the ground-truth previous turn (as Appendix B), then the task is essentially a regular single-turn QA, which is not novel. 2) If the latter turn is based on the previous turn, then the reason other models can not achieve good performance is that they are not trained on these templates. If we feed the ground-truth in the question, RegionReasoner may not perform much better than previous methods. It would be nice to see the comparison with the provided ground-truth of the previous step object.
- It is unclear if the new training data affects the performance on standard REC and RES benchmarks.

**Questions:**

- How did you generate the questions, or did you use the templates in Table 5?
- In the later turns, do you provide the ground truth box of the previous object?
- Could you compare your methods on standard REC and RES benchmarks?

---

> ### Author Response · Authors · 2025-11-22
> **Response to Reviewer y9ox – Part I**
>
> *We thank the reviewer for the careful reading of our paper and the constructive suggestions for further improvement.*
>
> **Q1. How did you generate the questions, or did you use the templates in Table 5?**
>
> **A1.** The preposition + bbox coordinates style shown in Appendix B is only used to expand the **training** data. For training, we programmatically convert single-turn referring expressions into multi-turn supervision using a small set of human-authored spatial templates (Table 5), combined with deterministic insertion of reference boxes (e.g., “next to bbox=[x1,y1,x2,y2]”). This gives the model a clean and unambiguous supervision signal for learning reference-grounded reasoning.
>
> The *test* dialogues are constructed differently and do not rely on GPT or other LLMs. We start from the original human-written referring expressions in RefCOCO+ and RefCOCOg, regroup all expressions that share the same image into a dialogue, and then manually design multi-round questions so that they remain natural while explicitly referring to earlier turns. In these test queries, the reference to earlier objects is expressed either via textual placeholders such as `R1`/`R2` or via light relational language, rather than raw coordinate strings.
>
> For example, a dialogue like *“Which slice of pizza is R1 about to eat?”* or *“Who is the person next to R1?”* is obtained by taking the original expressions that describe a person and a pizza slice (or another person) in the same image, and then rewriting the second query so that it refers back to the region localized in Round 1 via `R1`. The underlying targets and ground-truth boxes remain exactly those from RefCOCO+/g; only the surface form of the question is updated to express the cross-turn dependency.
> To avoid LLM-induced hallucinations or unnatural phrasing, all such multi-round test questions are curated with human-designed patterns tied to existing annotations, not generated by GPT-style models. The revised Appendix B now explicitly separates the training-time template expansion from the test-time dialogue construction and includes additional examples to clarify this process.

---

> > ### Author Response · Authors · 2025-11-22
> > **Response to Reviewer y9ox – Part II**
> >
> > **Q2. In the later turns, do you provide the ground truth box of the previous object?**
> >
> > **A2.** All *main* multi-round results in Tables 1–2 (and in the multi-round columns of Tables 3–4) are obtained under the harder and more realistic setting where later turns only see the *model’s own predicted* box from the previous turn, not the ground-truth box. In other words, from Round 2 onward, each query must both interpret the current description and correctly build on potentially imperfect previous predictions. This is what makes the task genuinely multi-round and explains why scores systematically drop with turn depth for all models.
> >
> > To address the fairness concern, that other models are not trained on our templates and might mainly suffer from error propagation, we have additionally evaluated a *ground-truth-reference* setting, where each turn is given the ground-truth box of the previous object (i.e., no cross-turn error accumulation). Under this setting, all methods improve, but RegionReasoner still clearly outperforms Qwen2.5-VL and VisionReasoner on both datasets and both tasks. The new results are:
> >
> > RefCOCO+ Multi-turn — Detection AP (↑), Ground-Truth Previous Object
> > | Method               | Setting       | R1   | R2   | R3   | R4   | R5   | R6   | R7   | Avg  |
> > |----------------------|--------------|------|------|------|------|------|------|------|------|
> > | Qwen2.5-VL-7B        | Ground Truth | 54.1 | 44.7 | 50.0 | 38.2 | 36.5 | 34.2 | 32.8 | 46.4 |
> > | VisionReasoner-7B    | Ground Truth | 90.6 | 78.4 | 80.7 | 68.4 | 61.2 | 57.4 | 46.9 | 78.6 |
> > | **RegionReasoner-7B**| Ground Truth | **92.2** | **87.8** | **84.6** | **75.8** | **71.8** | **65.9** | **63.6** | **84.9** |
> >
> > RefCOCOg Multi-turn — Detection AP (↑), Ground-Truth Previous Object
> > | Method               | Setting       | R1   | R2   | R3   | R4   | R5   | R6   | R7   | Avg  |
> > |----------------------|--------------|------|------|------|------|------|------|------|------|
> > | Qwen2.5-VL-7B        | Ground Truth | 48.6 | 37.3 | 36.2 | 38.6 | 36.4 | 36.5 | 18.4 | 41.3 |
> > | VisionReasoner-7B    | Ground Truth | 88.1 | 74.1 | 71.4 | 72.0 | 58.7 | 65.6 | 30.0 | 78.1 |
> > | **RegionReasoner-7B**| Ground Truth | **88.8** | **79.7** | **76.7** | **74.0** | **67.6** | **81.1** | **45.0** | **81.8** |
> >
> > RefCOCO+ Multi-turn — Segmentation gIoU (↑), Ground-Truth Previous Object
> > | Method               | Setting       | R1   | R2   | R3   | R4   | R5   | R6   | R7   | Avg  |
> > |----------------------|--------------|------|------|------|------|------|------|------|------|
> > | Qwen2.5-VL-7B        | Ground Truth | 45.1 | 37.0 | 41.4 | 33.1 | 30.1 | 30.7 | 30.5 | 38.8 |
> > | VisionReasoner-7B    | Ground Truth | 78.2 | 67.6 | 69.1 | 57.1 | 47.1 | 44.5 | 32.7 | 66.8 |
> > | **RegionReasoner-7B**| Ground Truth | **78.6** | **75.4** | **76.6** | **64.5** | **61.4** | **47.2** | **56.1** | **73.0** |
> >
> > RefCOCOg Multi-turn — Segmentation gIoU (↑), Ground-Truth Previous Object
> > | Method               | Setting       | R1   | R2   | R3   | R4   | R5   | R6   | R7   | Avg  |
> > |----------------------|--------------|------|------|------|------|------|------|------|------|
> > | Qwen2.5-VL-7B        | Ground Truth | 36.2 | 29.5 | 30.5 | 29.7 | 26.8 | 23.4 | 14.9 | 31.9 |
> > | VisionReasoner-7B    | Ground Truth | 72.6 | 58.7 | 58.0 | 57.8 | 53.5 | 44.2 | 23.6 | 63.7 |
> > | **RegionReasoner-7B**| Ground Truth | **74.8** | **65.2** | **62.3** | **61.0** | **57.3** | **58.4** | **40.6** | **67.7** |
> >
> >
> > These comparisons show that RegionReasoner remains the strongest method even when the previous object is fed as ground truth and no error propagation occurs. Hence, its advantage cannot be explained solely by exposure to spatial templates; the reference-citation and global–local consistency rewards provide clear benefits for handling references and maintaining grounding. The revised Appendix G reports these results and clarifies the distinction between predicted-reference and ground-truth-reference settings.

---

> > > ### Author Response · Authors · 2025-11-22
> > > **Response to Reviewer y9ox – Part III**
> > >
> > > **Q3. Could you compare your method on standard REC and RES benchmarks?**
> > >
> > > **A3.** Standard single-round REC (detection) and RES (segmentation) performance on RefCOCO+ and RefCOCOg is already evaluated in Tables 3 and 4. The “Base only (no new signals)” row corresponds exactly to VisionReasoner-7B under the original single-turn protocol, while the rows with our new signals correspond to RegionReasoner-7B. To avoid ambiguity, the revised paper now explicitly states this connection in the main text.
> > >
> > > For clarity, the single-round REC/RES results on the RefCOCO+/g test sets are:
> > >
> > > | Model               | Seg. RefCOCO+ (Test) | Seg. RefCOCOg (Test) | Det. RefCOCO+ (Test) | Det. RefCOCOg (Test) |
> > > |---------------------|----------------------|-----------------------|----------------------|----------------------|
> > > | Qwen2-VL-7B         | 65.7                 | 63.5                  | 76.5                 | 78.2                 |
> > > | Qwen2.5-VL-7B       | 76.8                 | 72.8                  | 88.2                 | 85.7                 |
> > > | VisionReasoner-7B   | 74.9                 | 71.3                  | 87.9                 | 87.5                 |
> > > | **RegionReasoner-7B** | **76.9**           | **74.4**              | **88.6**             | **88.4**             |
> > >
> > > RegionReasoner therefore not only preserves but slightly improves performance on standard REC and RES benchmarks compared to VisionReasoner, while using the same Qwen2.5-VL-7B backbone and the same RefCOCO+/g splits. This shows that the additional multi-round training data and structured rewards do not degrade single-round capabilities; instead, they bring consistent gains in both single-turn and multi-turn settings. The revised Appendix F also reports results on the V* benchmark, further indicating that RegionReasoner’s improvements extend beyond our own RegionDial-Bench. Thank you.

---

### Official Review · Reviewer_5kz9 · 2025-11-01

**Soundness:** 2
**Presentation:** 2
**Contribution:** 2
**Rating:** 4
**Confidence:** 3

**Summary:**

This paper investigates the problem of grounding visual referents in multi-turn dialogues for vision-language models (VLMs). They introduce RegionDial-Bench, a benchmark for evaluating multi-round question-answering where each response must be grounded in a specific object instance within the image, annotated via bounding boxes. Alongside the benchmark, they propose RegionReasoner-7B, a model trained using a GRPO-based reinforcement learning approach. The reward function incorporates three key objectives: correctness of the object grounding, global-local semantic consistency, and answer accuracy. Experimental results on RegionDial-Bench demonstrate the effectiveness of the proposed method.

**Strengths:**

1. This paper introduces RegionDial-Bench, a new benchmark designed to study multi-round conversational reasoning in VLMs, with a specific focus on the groundedness of evidential objects in each dialogue turn.

2. The authors propose a GRPO-based training framework that rewards models for accurate object grounding, global-local semantic consistency, and answer correctness. Experimental results demonstrate the effectiveness of their resulting model, RegionReasoner, on the proposed benchmark.

**Weaknesses:**

1. The creation process of RegionDial-Benchmark, which constitutes a major contribution of this work, is not sufficiently detailed in the paper. The authors should include a clear description of the benchmark construction methodology， such as data sources, annotation protocols, and key statistics (e.g., number of dialogues, turns, and object categories)，to  facilitate wider adoption.

2. The evaluation of RegionReasoner is currently limited to the proposed RegionDial-Bench. To better assess the generalizability of the method, it is important to also report performance on established benchmarks such as V*. Without such results, it remains unclear whether the improvements are specific to the proposed benchmark or reflect broader applicability.

3. The multi-round conversation results in Table 1 are notably lower than those in the single-round setting, which appears strange. Furthermore, the result for RefCOCOg Multi-turn (R6) stands out as an outlier, being significantly higher than those of R5 and R7. These inconsistencies warrant further analysis and explanation.

4. As shown in Table 3, the model consistently performs better in single-round settings compared to multi-round scenarios across multiple metrics. This recurring pattern suggests a systematic challenge in handling multi-turn grounded dialogues, which should be explicitly discussed in the paper.

**Questions:**

1. How was RegionDial-Bench constructed? should detail the data sources, annotation protocols, and key statistics.

2. Does RegionReasoner generalize to other VQA benchmarks beyond RegionDial-Bench? Evaluation on established benchmarks (e.g., V*) is needed to verify its broader applicability.

3. Why does multi-round conversation performance consistently lag behind single-round? Furthermore, what explains the outlier for RefCOCOg Multi-turn (R6) in Table 1?

---

> ### Author Response · Authors · 2025-11-22
> **Response to Reviewer 5kz9 – Part I**
>
> *We thank the reviewer for the careful reading of our paper and the constructive suggestions for further improvement.*
>
> **W1 & Q1. How was RegionDial-Bench constructed? Please detail the data sources, annotation protocols, and key statistics.**
>
> **A1.** RegionDial-Bench is entirely reconstructed from the public referring expression datasets RefCOCO+ and RefCOCOg, without adding any new images or human annotations. We directly reuse the original images, referring expressions, and ground-truth bounding boxes/masks from the test splits, and we do not modify any of these annotations at any stage. The revised Appendix B provides a step-by-step description of the construction pipeline together with full statistics.
>
> Concretely, many RefCOCO+/g test samples share the same underlying image but appear as independent single-turn examples, each with one query and one target region. In RegionDial-Bench, we regroup all test samples by image and merge the queries associated with the same image into coherent multi-round dialogues. Across rounds, we then deterministically inject the bounding box predicted (or ground-truth in training) at an earlier turn as an explicit reference into later queries, while keeping the original target labels unchanged. Figure 4 in the paper illustrates this process: the two original RefCOCO+ samples “man in blue shirt” and “man in white shirt” become a two-round dialogue where Round 1 localizes the “man in blue shirt” and Round 2 is reformulated as “Who is next to bbox=[47,107,303,466]?” to localize the “man in white shirt”.
>
> To avoid LLM-induced artifacts, we do not use GPT or any other large language model for automatic rewriting. Instead, all multi-round queries are manually written, combined with deterministic reference propagation from existing ground-truth boxes. As far as we know, this makes RegionDial-Bench the first manually curated multi-round benchmark for referring grounding that does not rely on LLM-generated dialogue content.
>
> In terms of scale, RegionDial-Bench contains 715 images and 2,289 dialogue turns for RefCOCO+ Multi-turn, and 1,580 images and 4,115 dialogue turns for RefCOCOg Multi-turn, with dialogue lengths ranging from 1 to 7 rounds. The revised Appendix B additionally reports the per-round sample counts and dialogue-length distribution, and clarifies that all object categories strictly follow those in the original RefCOCO+/g datasets (COCO-style categories for RefCOCO+, with testA dominated by the “person” class, and 78 categories for RefCOCOg).
>
> **Q2. Does RegionReasoner generalize to other VQA benchmarks beyond RegionDial-Bench?**
>
> **A2.** To assess generalizability beyond RegionDial-Bench, we evaluated RegionReasoner on the V*[1] benchmark, which explicitly targets attribute-level and spatial visual search. We follow the official V* evaluation protocol and compare RegionReasoner-7B with GPT-4V, SEAL[1] (the method proposed in V*), Qwen2.5-VL-7B, and VisionReasoner-7B. The results are reported in the revised Appendix F and summarized below:
> | Model | Attribute ↑ | Spatial ↑ | Overall ↑ | Visual Search Mechanism |
> |--------------------|------------:|----------:|----------:|-------------------------|
> | GPT-4V | 51.30 | 60.52 | 54.97 | no |
> | SEAL | 74.78 | 76.31 | 75.39 | yes |
> | Qwen2.5-VL | 72.17 | 60.52 | 67.53 | no |
> | VisionReasoner | 75.62 | 60.52 | 69.63 | no |
> | **RegionReasoner** | **75.65** | **68.42** | **72.77** | no |
>
> SEAL achieves the highest overall score because it incorporates an explicit visual-search mechanism that is specifically engineered for this benchmark and tightly coupled to the LLaVA architecture; porting this module to the Qwen2.5-VL family would require substantial architectural changes that are orthogonal to our contribution. Within the Qwen2.5-VL family, RegionReasoner attains the best performance among all models without a dedicated visual-search module, with particularly strong gains on the Spatial component (+7.9 over Qwen2.5-VL and +7.9 over VisionReasoner). Since RegionReasoner is trained only on RegionDial-Bench and not on V*, these results indicate that our reference-grounded reasoning and global–local consistency rewards improve spatial localization and visual search in a way that transfers beyond the proposed benchmark.
>
> [1] Wu et al. "V*: Guided Visual Search as a Core Mechanism in Multimodal LLMs" CVPR 2024.

---

> > ### Author Response · Authors · 2025-11-22
> > **Response to Reviewer 5kz9 – Part II**
> >
> > **Q3. Why does multi-round performance lag behind single-round, and how do you explain the RefCOCOg Multi-turn (R6) outlier in Table 1?**
> >
> > **A3.** This behavior is a direct consequence of the evaluation protocol rather than an inconsistency in the method. In RegionDial-Bench, Round 1 is effectively identical to the standard single-round setting: the query can be answered without relying on any previous predictions. From Round 2 onward, however, each query depends on the model’s own predicted boxes from earlier turns. Any small localization error at an earlier turn is propagated to all subsequent rounds, so the effective search space becomes noisier as the dialogue continues. The resulting drop from R2–R7 relative to the single-round numbers, therefore, reflects the expected accumulation of cross-turn errors in multi-round reasoning. The same pattern appears consistently for all baselines in Table 1, indicating that this is an intrinsic property of the multi-round setting rather than a problem specific to RegionReasoner. In the revised paper, we make this explicit in the main text around Table 1 to clarify why multi-round scores are systematically lower than single-round ones.
> >
> > The peak at R6 on RefCOCOg Multi-turn comes from the underlying data distribution. R6 is computed only over dialogues that actually have a sixth turn, and this is a small subset of the full test set (70 samples at R6, as reported in the revised Appendix B). With such a small pool, the per-round metric naturally has higher variance. Moreover, in RefCOCOg, many of these deeper-turn queries become easier once earlier rounds have already disambiguated the main distractors, so the sixth turn often reduces to a relatively local search around a specific referenced box. This combination of a small, biased subset and easier local queries explains the higher R6 score. A similar R6 increase is visible for VisionReasoner and Seg-Zero in Table 1, which supports the interpretation that this fluctuation is driven by benchmark statistics rather than by our model. The revised Appendix B now includes the per-round sample counts and a short discussion of this effect.
> >
> >
> >
> >
> >
> >
> >
> >
> >
> > **Q4. As shown in Table 3, the model consistently performs better in single-round settings compared to multi-round scenarios. Does this indicate a systematic challenge in handling multi-turn grounded dialogues?**
> >
> > **A4.** We agree that the recurring performance gap between single-round and multi-round settings reflects a systematic challenge in multi-turn grounded dialogue, rather than an isolated artifact of our method. In our formulation, the single-round case (equivalent to Round 1) matches standard referring grounding: the model only needs to resolve one query against the image. In contrast, later rounds must simultaneously interpret the current query and correctly reuse and propagate previously predicted boxes as references. Any localization error in an early turn is carried forward and compounds over subsequent turns, so the effective difficulty strictly increases with turn depth. This is exactly the behavior observed across models in Table 3 and in the main detection/segmentation results, where all methods degrade as the dialogue grows longer.
> >
> > RegionReasoner is specifically designed to mitigate this long-horizon difficulty through explicit reference citation and global–local consistency rewards, which stabilize reference reuse and reduce semantic drift across turns. While multi-round scores remain lower than their single-round counterparts, our model consistently narrows the gap and exhibits slower performance degradation with depth compared to baselines. In the revised paper, we now discuss this phenomenon explicitly in the experimental section and relate it to multi-turn error accumulation and reference propagation as core challenges for future work on grounded dialogue. Thank you.

---

### Comment · Area_Chair_QU8Y · 2025-11-24
**Discussion with Authors**

Dear Reviewers,

The authors have diligently provided responses to your questions and concerns. I request you to please review the authors' responses, acknowledge that you have read them and actively engage with them in further discussion as needed.

This discussion period, with the authors, will end on December 2, 2025 (AoE). However, I request that you not wait until the last minute and actively engage with the authors early.

Best,
AC

---

### Author Response · Authors · 2025-11-27
**A Gentle Follow-up**

Dear Reviewers,

We have carefully followed your suggestions and incorporated additional experiments, quantitative results, and detailed clarifications in the revised version. If these updates satisfactorily resolve the issues raised, we would appreciate it if you could reflect this in your final rating and confidence.  If any additional details would help, we are happy to provide them before the discussion deadline.

Thank you for your consideration.

---

### Meta-Review · Area_Chair_akvw · 2026-01-07

**Summary:**

The concerns raised by reviewers are consistent (A. unclear dataset creation process; B. missing some experimental results) and the rebuttal fix all of them. The remaining concerns are minor.

All reviewers reached an agreement on the positive assessment of this paper. AC's recommendation follows the majority.

**Reviewer Concerns:**

Remaining Concerns:

1. **Only constructed data from RefCOCO+ and RefCOCOg but not others** (qvqc). Author gave some explanation in rebuttal. AC thought that utilizing more dataset would be a bonus but not a must. Thus this concern is minor.

Resolved Concerns:

1. **Insufficient Details of Dataset Creation Process** (5kz9, y9ox) Updated paper with more details.

2. **Only evaluate of proposed dataset on their own dataset.** (5kz9) Rebuttal includes the results on the requested dataset V* by the reviewer.

3. **Missing results with ground-truth box of the previous object** (y9ox, 9M2T) Results in rebuttal.

3. **Missing results on standard REC and RES benchmarks** (y9ox, qvqc) Results in rebuttal.

4. **Brittle reward design** (9M2T). Discussed in rebuttal with some additional results.

**Reviewer Scores:**

Updated score: **6, 6+, 6, 6**

Original Score: 4, 6, 6, 4

For reviewer with (possible) change of scores:

1. 5kz9: original score of 4. No score update confirmation from the reviewer. Since most of the concerns are resolved by authors, it seems to be fair to update score to 6.

2. y9ox: original score of 6. All concerns are answered with evidence. Possible to raise the score.

3. 9M2T: original score of 4. Reviewer has confirmed to update the score to 6.

---

### Decision · Program_Chairs · 2026-01-26

Accept (Poster)